# *Mayday* sustains trans-synaptic BMP signaling required for synaptic maintenance with age

Jessica M Sidisky, Daniel Weaver, Sarrah Hussain, Meryem Okumus, Russell Caratenuto, Daniel Babcock*

Department of Biological Sciences, Lehigh University, Bethlehem, United States

**Abstract** Maintaining synaptic structure and function over time is vital for overall nervous system function and survival. The processes that underly synaptic development are well understood. However, the mechanisms responsible for sustaining synapses throughout the lifespan of an organism are poorly understood. Here, we demonstrate that a previously uncharacterized gene, *CG31475*, regulates synaptic maintenance in adult *Drosophila* NMJs. We named *CG31475 mayday* due to the progressive loss of flight ability and synapse architecture with age. Mayday is functionally homologous to the human protein Cab45, which sorts secretory cargo from the Trans Golgi Network (TGN). We find that Mayday is required to maintain trans-synaptic BMP signaling at adult NMJs in order to sustain proper synaptic structure and function. Finally, we show that mutations in *mayday* result in the loss of both presynaptic motor neurons as well as postsynaptic muscles, highlighting the importance of maintaining synaptic integrity for cell viability.

## Introduction

Synaptic communication is key for proper nervous system function. Through developmental studies, we have learned that synaptic communication is first established by coordinated events between the presynaptic neuron and the postsynaptic cell (*Collins and DiAntonio, 2007*; *Turrigiano and Nelson, 2004*). By contrast, how these structures are maintained over time is poorly understood. The importance of synaptic maintenance has been highlighted by recent evidence implicating synapse dysfunction in aging as well as during the early stages of neurodegenerative diseases such as Amyotrophic Lateral Sclerosis (ALS), Alzheimer's Disease (AD), and Parkinson's Disease (PD) (*Lodato et al., 2018*; *López-Erauskin et al., 2018*; *López-Otín et al., 2013*; *Munsie et al., 2015*; *Oddo et al., 2003*; *Selkoe, 2002*). Thus, understanding the mechanisms by which synaptic integrity is maintained is crucial for addressing synaptic dysfunction.

One of the most well-characterized structures used to study synaptic function in both vertebrates and invertebrates is the neuromuscular junction (NMJ) (*Broadie and Bate, 1993a*; *Broadie and Bate, 1993b*; *Sanes and Lichtman, 1999*). The *Drosophila* NMJ provides a genetically tractable model system to examine processes that are well conserved across species (*Featherstone and Broadie, 2000*; *Keshishian et al., 1996*). The *Drosophila* larval NMJ in particular has been fundamental to our understanding of synaptic transmission and the growth and development of synaptic structure and function (*Collins and DiAntonio, 2007*; *Harris and Littleton, 2015*). However, larval NMJs are transient structures that exist for only a few days before being dismantled during metamorphosis (*Liu et al., 2010*; *Tissot and Stocker, 2000*), limiting their utility in identifying the mechanisms responsible for maintaining synaptic integrity with age.

Among the most prominent neuromuscular synapses in adult *Drosophila* are those of the indirect flight muscles (*Costello and Wyman, 1986*; *Fernandes et al., 1991*; *Fernandes et al., 1996*; *Fernandes and Keshishian, 1996*; *Fernandes and Keshishian, 1998*; *Hebbar and Fernandes,*

*For correspondence:
dab416@lehigh.edu

Competing interests: The authors declare that no competing interests exist.

*2004*; *Hebbar and Fernandes, 2005*). One set of IFMs, the Dorsal Longitudinal Muscles (DLMs), are composed of six large muscle fibers innervated by five motor neurons on each side of the thorax (*Fernandes et al., 1991*; *Fernandes and VijayRaghavan, 1993*; *Hebbar and Fernandes, 2004*; *Shafiq, 1963*; *Shafiq, 1964*; *Takahashi et al., 1970*). Once the DLM NMJs are established, these stable structures are present throughout the lifespan of the organism (*Danjo et al., 2011*; *Fernandes and VijayRaghavan, 1993*; *Fernandes and Keshishian, 1998*; *Hebbar and Fernandes, 2004*; *Truman, 1990*). These NMJs are part of the Giant Fiber (GF) pathway that propels flight behavior (*Allen et al., 2006*). Thus, we can monitor the activity of DLMs by assaying flight behavior as a read-out of synaptic integrity (*Babcock and Ganetzky, 2014*; *Benzer, 1973*; *Deak, 1977*; *Dudai et al., 1976*; *Thomas and Wyman, 1984*). Additionally, the DLM NMJs form a tripartite synapse composed of a presynaptic motor neuron, postsynaptic muscle cell, and associated glial cell, that provide the ability to understand synaptic function at the cellular and molecular level (*Danjo et al., 2011*). This model also allows for expression of transgenes in non-essential tissue, particularly the DLM motor neurons that are easily accessible (*Coggshall, 1978*; *Danjo et al., 2011*; *Godenschwege et al., 2006*; *Hebbar and Fernandes, 2004*). Together, we can assess the morphological and functional properties of adult DLM NMJs to elucidate the mechanisms responsible for sustaining synapses in aging adults, then apply this to understand how synapses deteriorate in neurodegenerative diseases.

Although we may not currently understand the processes involved in maintaining synaptic structure and function, there are a few key pathways that are crucial for regulating synaptic growth, organization and stability during synaptic development (*Ball et al., 2010*; *Ballard et al., 2010*; *McCabe et al., 2003*; *Packard et al., 2002*). Specifically, in *Drosophila* one key signaling cascade that involves coordination between the presynaptic motor neurons and postsynaptic muscle cells is the bone morphogenic protein (BMP) signaling cascade (*Aberle et al., 2002*; *Marqués et al., 2002*; *McCabe et al., 2004*; *McCabe et al., 2003*; *Rawson et al., 2003*; *Sweeney and Davis, 2002*). The morphogen glass bottom boat (Gbb), the *Drosophila* ortholog to mammalian BMP7, is secreted in a retrograde manner from the postsynaptic muscle cell to the presynaptic motor neuron (*Chen et al., 1998*; *McCabe et al., 2003*; *Wharton et al., 1999*; *Wharton et al., 1991*). Currently, it is not understood how this pathway could function past development. This suggests that this signaling cascade could play a role in maintaining synaptic integrity.

Gaining a better understanding of synaptic dysfunction should help to identify strategies involved in maintaining synaptic integrity with age. Here, we identify Mayday, a resident Golgi protein that is required to maintain trans-synaptic signaling across adult NMJs. We find that mutations in *mayday* impair retrograde BMP signaling, resulting in degradation of synaptic structure and function. Finally, we demonstrate that this sustained trans-synaptic signaling is required to maintain the viability of both presynaptic motor neurons and postsynaptic muscles.

## Results

### Progressive loss of flight in *3PM71* mutants

Maintaining synaptic communication throughout the lifespan of an organism is crucial for nervous system function and survival. To better understand how these structures are sustained beyond development, we carried out a forward genetic screen assaying locomotor behavior in adult *Drosophila* using a collection of temperature-sensitive paralytic mutants associated with neurodegeneration (*Palladino et al., 2002*). We utilized a high-throughput flight assay to screen for the progressive loss of flight ability with age (*Babcock and Ganetzky, 2014*). Flight behavior in *Drosophila* is powered primarily by the indirect flight muscles (IFMs). Among these muscles are the dorsal longitudinal muscles (DLMs) which are an important component of the Giant Fiber escape response (*Allen et al., 2006*). The DLMs are composed of six large muscle fibers labeled from a to f (dorsal to ventral) and are innervated by five motor neurons, forming the NMJs (*Figure 1A–C*). These motor neurons are located in the thoracic ganglion, and send their axons into the nearby DLMs (*Figure 1D–E*).

In our forward genetic screen, we isolated *3PM71*, a mutant that demonstrated progressive loss of flight. At Day 3, both wild type (WT) and *3PM71* mutants performed similarly, with average landing heights of 77.6 and 74.1 cm. While WT flies continued to perform well at day 25, however,

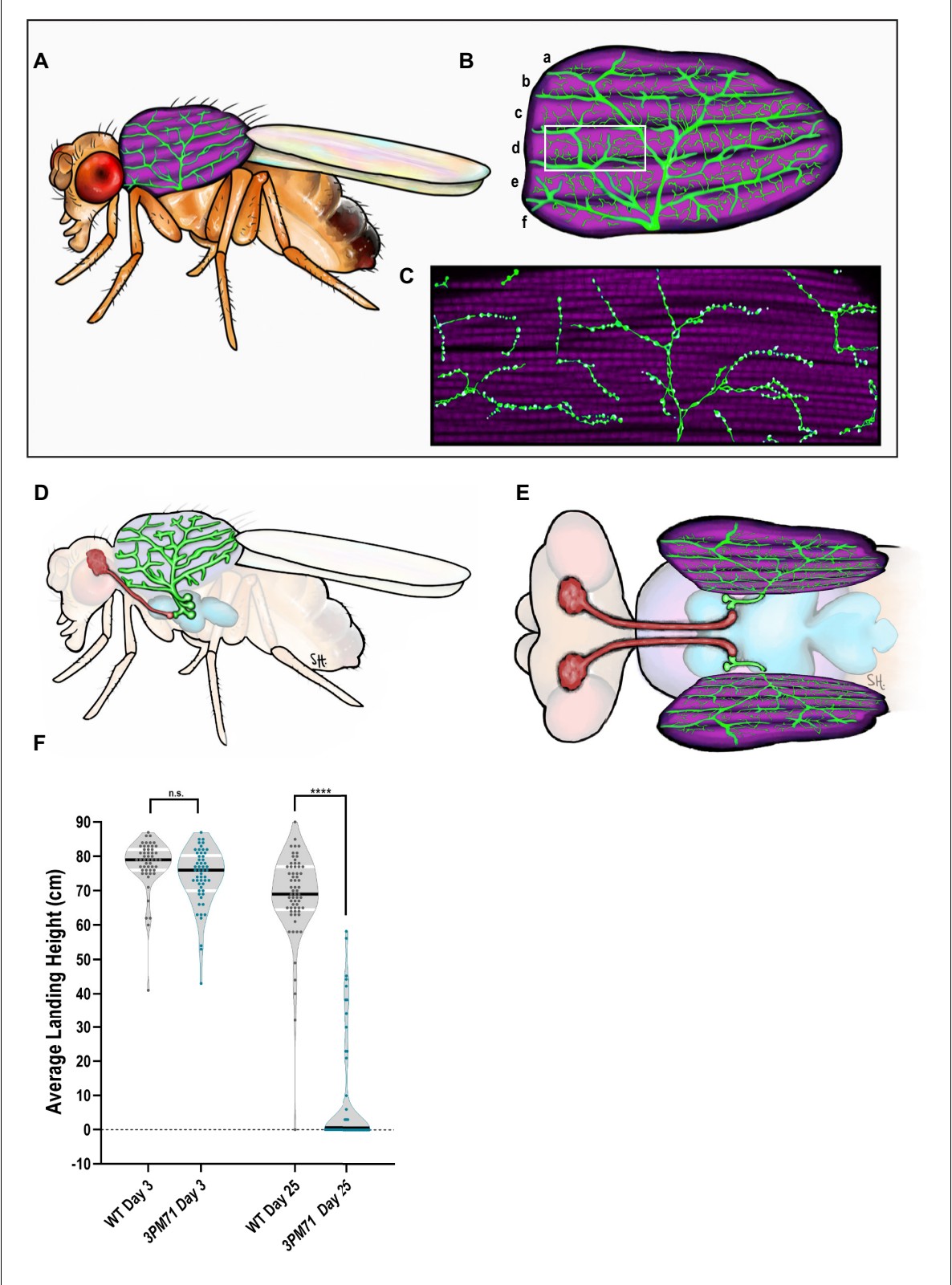

**Figure 1.** Progressive loss of flight in *3PM71* mutants. (**A**) Illustration depicting the location and morphology of the dorsal longitudinal muscles within the thorax. (**B**) The six muscle fibers and five motor neurons that innervate them are highlighted. (**C**) Further magnification of a single muscle fiber reveals the en passant boutons and NMJ structures. (**D**) Side and (**E**) Dorsal view of motor neuron cell bodies located within the thoracic ganglion. (**F**) The average landing height of Wild Type (WT) and *3PM71* mutants at days 3 and 25. Each dot represents a single fly in the violin plot. Sample sizes for

*Figure 1 continued on next page*

*Figure 1 continued*

each plot: WT Day 3 n = 51, with a landing average of 77.6 cm, *3PM71* Day 3 n = 54, landing average 74.1 cm, Day 25 WT n = 65, landing average 68.3 cm, and *3PM71* Day 25 n = 51, and landing average 9.4 cm. *3PM71* mutants have a progressive loss of flight ability in comparison to WT. Black bars represent median values. ****, p-value<0.0001 using A Brown-Forsythe and Welch ANOVA tests with Post hoc Games-Howell's multiple comparisons. N.S. = Not Significant.

The online version of this article includes the following source data for figure 1:

**Source data 1.** Raw data for *Figure 1* flight scores.

*3PM71* mutants had a significantly lower landing height, suggesting a progressive loss of locomotor ability (*Figure 1F*).

To determine whether the structural integrity of NMJs is compromised along with the progressive flight defects seen in in *3PM71* mutants, we assessed the gross morphology of NMJs at both early and late time points. DLMs were stained using horseradish peroxidase (HRP) to stain motor neurons. At an early age, WT and *3PM71* mutants had comparable motor neuron and muscle morphology (*Figure 2A–B*). At day 25, however, motor neuron integrity is severely compromised in *3PM71* mutants, with little HRP staining remaining relative to WT (*Figure 2C–D*). Between day 3 and day 25, 3PM71 mutants show a significant decrease in total branch length, branch number, and bouton number (*Figure 2E–G*). These data support the idea that progressive loss of flight in *3PM71* mutants is caused by denervation of DLM NMJs. Thus, the gene(s) responsible for the *3PM71* mutant phenotype likely play a neuroprotective role in maintaining synaptic structure and function.

## *3PM71* mutation maps to *CG31475*

To elucidate the gene(s) responsible for the progressive loss of flight in *3PM71* mutants, we first mapped this mutation to the third chromosome and used deficiency stocks to identify the location of the mutation on this chromosome. *3PM71* is a recessive mutation, as heterozygotes have a comparable flight performance as controls. When a single copy of *3PM71* is combined with *Df(3R) ED5938*, flies had a loss of flight comparable to *3PM71* recessive mutants (*Figure 3A*), suggesting that the gene responsible for the flight defect lies within this uncovered region.

To further identify the specific genetic lesion, we crossed mutants of genes within this region to *3PM71* and measured progressive flight performance. This analysis included three transposon insertion alleles of the previously uncharacterized gene *CG31475*, including *CG31475^MI08666* (*Nagarkar-Jaiswal et al., 2015*), *CG31475^MI08258* (*Nagarkar-Jaiswal et al., 2015*), and *CG31475^MB03509* (*Bellen et al., 2011*). Flies bearing each of these alleles flew well when crossed to WT. Interestingly, two of the three alleles displayed a significant flight defect when tested along with *3PM71*, and all three alleles phenocopied *3PM71* when tested in combination with the deficiency (*Figure 3B*). These results strongly support the idea that the mutation responsible for the NMJ defects in *3PM71* maps to *CG31475*.

We next performed DNA sequencing of the *CG31475* gene region in both WT and *3PM71* flies to reveal the specific nature of the *3PM71* mutation. In comparison to WT, *3PM71* mutants harbor a deletion and an amino acid change within the second exon of *CG31475* (*Figure 3C*). As this gene has remained previously uncharacterized, we now refer to *CG31475* as *Mayday (Myd)* due to its role in the progressive loss of flight performance. We also now refer to the *3PM71* allele as *myd^3PM71*.

Finally, we performed qPCR to assess transcript levels of *myd* in each of the alleles assessed in this study. Interestingly, transcript levels of *myd* were not significantly altered in these alleles in comparison to WT flies (*Figure 3—figure supplement 1*). Thus, while transcript levels do not appear to be increased or decreased, it is possible that the flight phenotypes caused by these mutations instead is a result of interfering with *myd* function.

## Mayday is required in muscle tissue to maintain synaptic integrity

To determine the tissue(s) in which Mayday is required to exert its function, we utilized the Gal4/UAS system to knock down Myd using RNA interference (RNAi) (*Dietzl et al., 2007*) using various tissue-specific drivers. We first knocked down Myd ubiquitously with Tubulin-Gal4 (Tub-Gal4) and found it to be lethal, with no adult progeny emerging (*Figure 4A*). The lethality of *myd* RNAi using Tub-Gal4 illustrates the essential role of *myd*. We also validated the specificity of the *UAS-myd^RNAi*

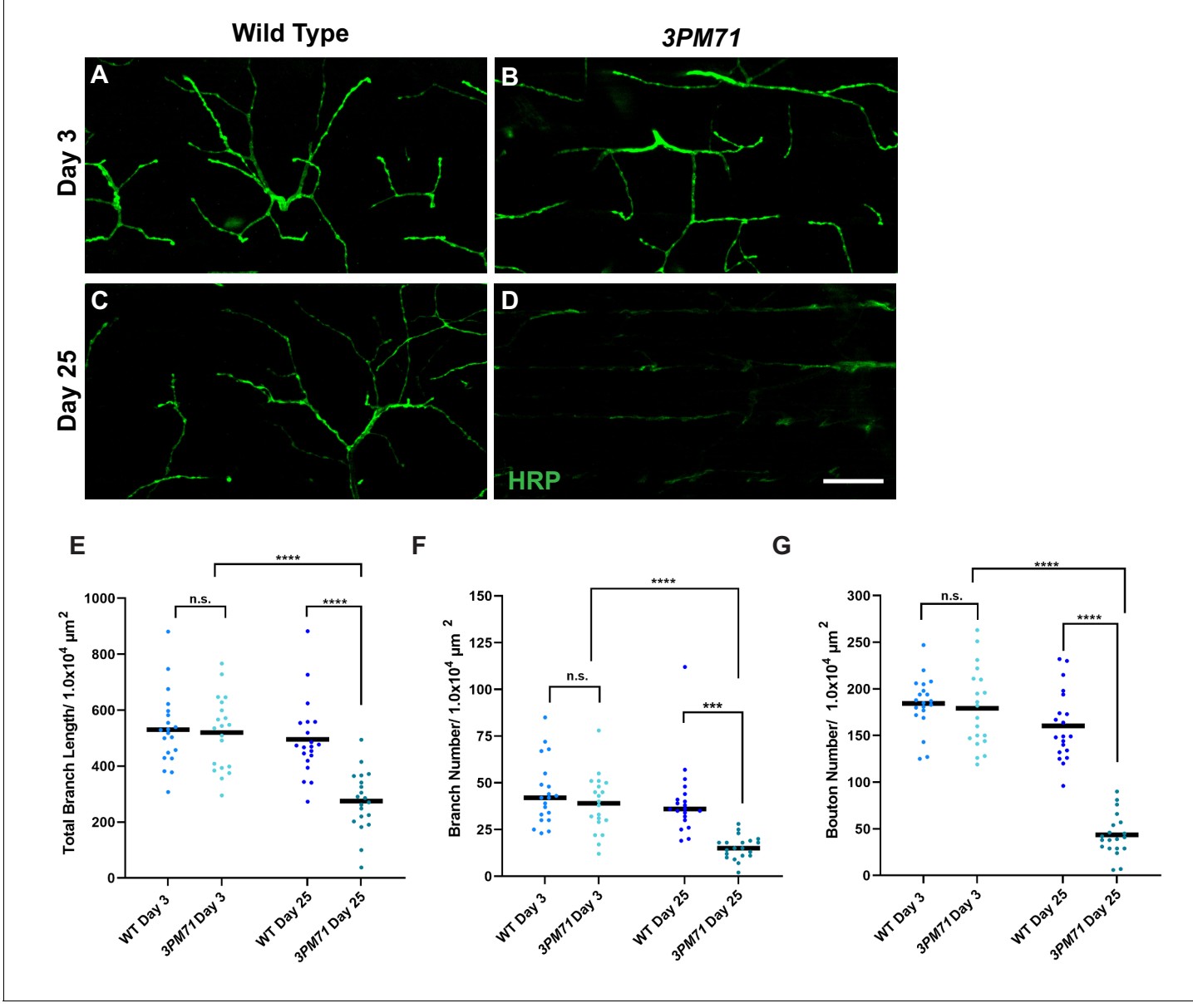

**Figure 2.** Progressive loss of NMJ integrity in *3PM71* mutants. (A–D) Confocal images of DLM NMJs labeled with FITC-conjugated HRP (green) at 63X magnification. (E–G) Quantification of (E) Total Branch Length, (F) Branch number, and (G) Bouton Number at both WT and *3PM71* NMJs. Sample size was n = 20 for each group. Black bars represent mean values. The mean total branch lengths for each data set from left to right as depicted in the graph (530.5 µM, 519.5 µM, 495.1 µM, and 274.9 µM). The mean branch number for each group as follows: 45, 39, 40, and 15. The mean bouton number (G) for each group as ordered in the panel, 185, 179, 161, and 43. ****, p-value<0.0001, *** p-value<0.001 using a Brown-Forsythe and Welch ANOVA tests with Post Hoc Dunnett's multiple comparisons. N.S. = Not Significant. Scale bar in D represents 20 µM for all confocal images.
The online version of this article includes the following source data for figure 2:

**Source data 1.** Raw data for synaptic morphology.

line by co-expressing a WT version of *UAS-myd*. The ability of *UAS-myd* to restore the RNAi pheno-type demonstrates the specificity of the RNAi construct (*Figure 4—figure supplement 1*). We next used Gal4 drivers that target tissues associated with the tripartite synapse, including neurons, glia, and muscle tissues (*Danjo et al., 2011*). When Myd is knocked down using a pan-neuronal driver (*Elav^{C155}-gal4*) or a motor neuron driver (*OK371-Gal4*), flight behavior phenocopied heterozygous controls. We found a similar result using a pan-glial driver (*Repo-Gal4*) as there was no progressive loss of flight (*Figure 4B–D*). Finally, we examined knockdown of *myd* in muscle tissue (*MHC-Gal4*)

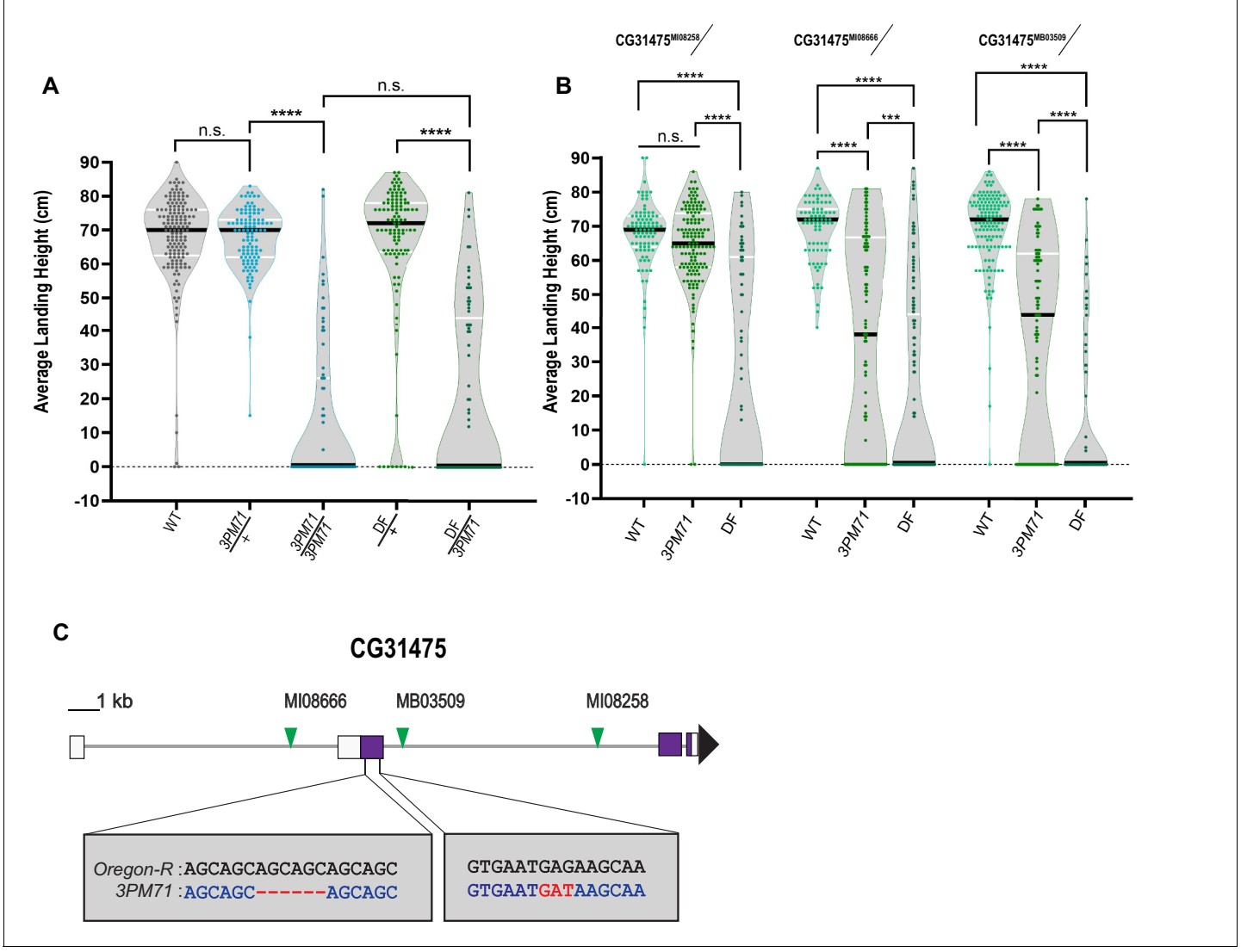

**Figure 3.** *3PM71* mutation maps to *CG31475*. (**A**) Deficiency mapping of *3PM71* mutants to a small region of chromosome 3R using Df(3R)ED5938. The average landing height (cm) and sample size for each group as ordered in the figure are, WT 67.4 cm, n = 149, *3PM71*/+ 67.4 cm, n = 108, *3PM71* 13.3 cm, n = 77, Df(3R)ED5938/+ 65.4 cm, n = 113, and Df(3R)ED5938/*3PM71* 20.2 cm, n = 73. (**B**) Flight performance of insertion alleles in *CG31475* (CG31475^MI08258, CG31475^MI08666, and CG31475^MB03509) alone and in combination with *3PM71* and Df(3R)ED5938. For each group, the landing average (cm) and sample size are as follows: CG31475^MI08258/+ 67.5 cm, n = 92, CG31475^MI08258/*3PM71* 64.3 cm, n = 140, CG31475^MI08258/Df 26.9 cm, n = 78, CG31475^MI08666/+ 69.3 cm, n = 81, CG31475^MI08666/*3PM71* 36.2 cm, n = 108, CG31475^MI08666/Df 20.4 cm, n = 156, CG31475^MB03509/+ 69.3 cm, n = 137, CG31475^MB03509/*3PM71* 36.9 cm, n = 101, and CG31475^MB03509/Df 11.6 cm, n = 72. (**C**) *3PM71* mutants map to the previously uncharacterized gene, CG31475. Green arrows represent the location of current CG31475 alleles. White and purple boxes represent exons. Black bars represent the median values. and ****p<0.0001, ***p<0.001, using A Brown-Forsythe and Welch ANOVA tests with Post hoc Games-Howell's multiple comparisons. N.S. = Not Significant.

The online version of this article includes the following source data and figure supplement(s) for figure 3:

**Source data 1.** Raw data for *Figure 3* flight scores.
**Source data 2.** Raw data for transcript expression.
**Figure supplement 1.** qPCR results.

and found a progressive loss of flight (*Figure 4E*), similar to the phenotype observed in *myd^3PM71* mutants. These data suggest that *myd* has a necessary role in postsynaptic muscle tissue in maintaining synaptic integrity.

To further validate the role of Myd in muscle tissue, we visualized the gross morphology of NMJs upon knocking down *myd* in muscles. MHC-Gal4 heterozygotes showed no obvious changes in

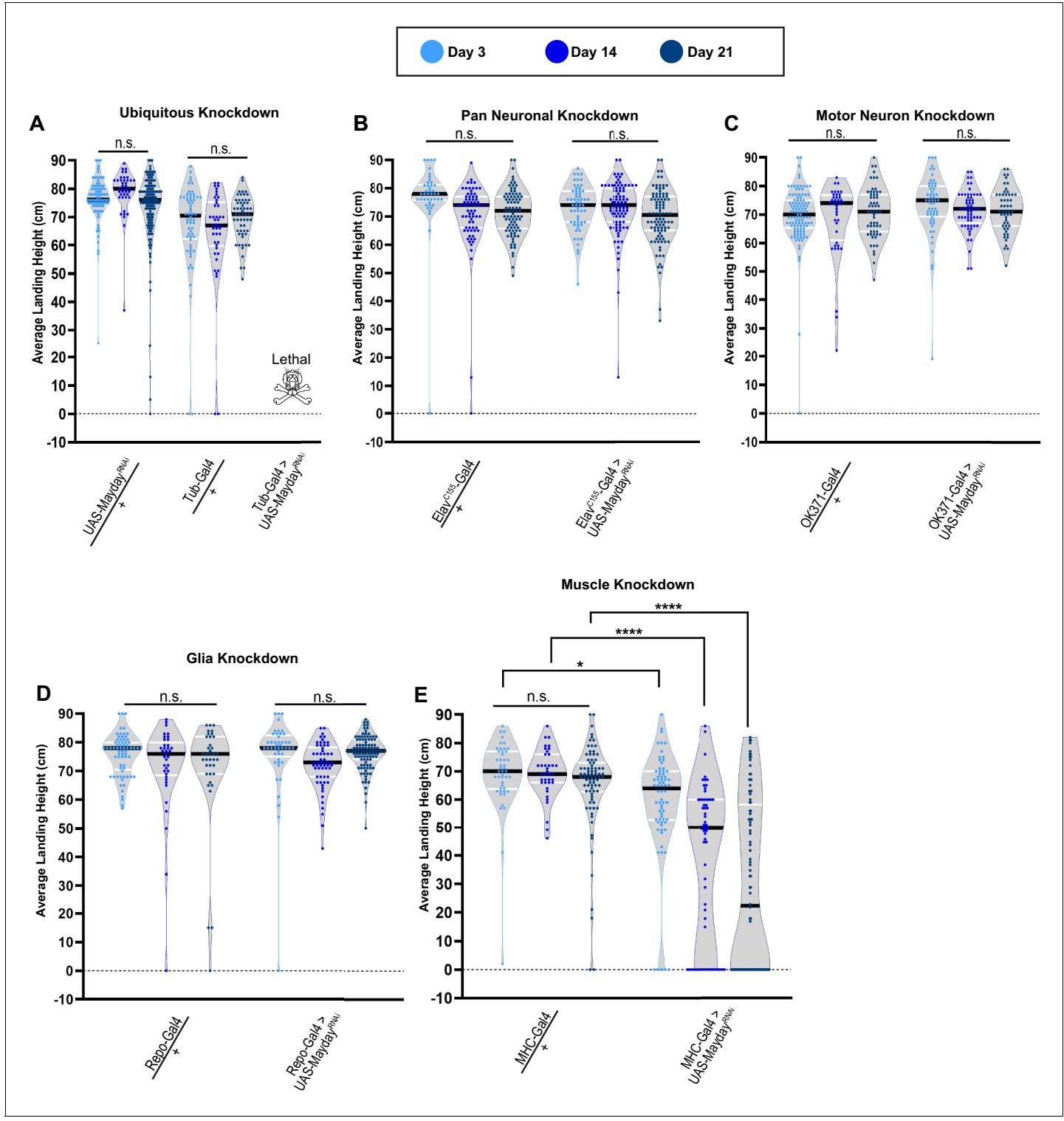

**Figure 4.** Mayday is required in postsynaptic muscle tissue. (A–E) Flight performance upon tissue-specific knockdown of *myd* with UAS-Dicer 2. Knockdown was assessed ubiquitously (A), pan-neuronally (B), in motor neurons (C), in glia (D), and in muscles (E). All lines used in the RNAi screen were also crossed to WT for controls. The average landing height and sample size for A-E are listed in **Supplementary file 1**. Black bars represent the median values. ****, p-value<0.0001, *, p-value<0.05, using A Brown-Forsythe and Welch ANOVA tests with Post hoc Games-Howell's multiple comparisons. N.S. = Not Significant.

The online version of this article includes the following source data and figure supplement(s) for figure 4:

**Source data 1.** Raw data for **Figure 4** flight scores.

*Figure 4 continued on next page*

*Figure 4 continued*

**Figure supplement 1.** Validation of *UAS-myd*^RNAi^ transgene.

**Figure supplement 1—source data 1.** Raw data for *Figure 4—figure supplement 1*.

**Figure supplement 2.** Morphology of Myd RNAi NMJs.

**Figure supplement 3.** Adult-specific knockdown of *UAS-myd*^RNAi^.

**Figure supplement 3—source data 1.** Raw data for *Figure 4—figure supplement 3*.

motor neuron or muscle integrity from day 3 through 21 (*Figure 4—figure supplement 2A–I*). At day 3, MHC-Gal4>*myd*^RNAi^ flies also showed NMJ integrity comparable to controls (*Figure 4—figure supplement 2J–L*). While muscle morphology remains normal at days 14 and 21, motor neuron morphology is severely decreased by day 14, and is almost entirely lost by day 21 (*Figure 4—figure supplement 2M–R*). Thus, the structural deterioration of NMJs in MHC-Gal4>*myd*^RNAi^ flies is consistent with the functional.

The finding that ubiquitous knockdown of *mayday* is lethal prior to adult eclosion suggests that this gene also has an important role in earlier stages of development. To test whether the defects in synaptic maintenance at adult NMJs are due to loss of Mayday function at earlier stages, we transiently knocked down *mayday* expression after adult eclosion and measured flight performance. Using a temperature-sensitive Gal80 (*McGuire et al., 2003*), flies were raised at 18°C until eclosion to prevent Gal4 expression. Upon eclosion, adult flies were shifted to 29°C to repress Gal80 and allow for the Gal4-mediated knockdown of *mayday*. Using both ubiquitous and muscle-specific knockdown, we found that flies still demonstrated a progressive loss of flight ability (*Figure 4—figure supplement 3*). Although *mayday* appears to have required roles throughout development, these results suggest that *mayday* is required throughout aging in order to maintain synaptic integrity.

## Mayday expression in muscle and motor neurons is necessary to maintain synaptic integrity

Given the loss of synaptic integrity in *myd*^3PM71^ mutants, we investigated the ability of wild-type Myd to prevent the progressive loss of flight. We tested for the restoration of flight behavior by expressing a wild-type Mayday construct bearing an N-Terminal Venus tag. Expression of UAS-venus::myd ubiquitously using Tub-Gal4 in a *Myd*^3PM71^ homozygous background rescued the progressive loss of flight observed in *Myd*^3PM71^ mutants, with a similar average landing height compared to wild-type flies (*Figure 5A*). These results further support the role of *myd* in maintaining synaptic integrity. Since muscle-specific knockdown of Myd using RNAi recapitulates the flight defect seen in *myd*^3PM71^ mutants, we next examined whether expressing UAS-venus::myd in muscle tissue in the *myd*^3PM71^ homozygous background could restore flight behavior. We found that muscle-specific expression was unable to restore flight behavior (*Figure 5A*), suggesting that expression of Myd in muscles is necessary, but not sufficient to maintain synaptic integrity. Although motor neuron-specific knockdown of Myd did not impair flight on its own, we next tested whether expression of WT Myd is also required in motor neurons to maintain synaptic integrity. By expressing Myd in presynaptic motor neurons and postsynaptic muscles simultaneously in a *Myd*^3PM71^ background, we found that flight ability was restored to levels comparable to WT controls as well as the ubiquitous rescue (*Figure 5A*). These results demonstrate that maintaining synaptic integrity requires both pre- and postsynaptic expression of Myd.

We also assessed whether expression of UAS-venus::myd could rescue the gross structural deficits in NMJ morphology in *myd*^3PM71^ mutants. We found that ubiquitous expression of UAS-venus::myd in the *Myd*^3PM71^ homozygous background was able to maintain HRP staining comparable to controls (*Figure 5B–D*), as well as total branch length, branch number, and bouton number (*Figure 5G–I*). Similar to our results with functional integrity, muscle-specific expression of UAS-venus::myd was unable to prevent NMJ deterioration as seen in *myd*^3PM71^ mutants (*Figure 5E and G–I*). However, expression of Myd in both pre-and post-synaptic tissue was able to restore these values (*Figure 5F–I*). Interestingly, the synapses in the pre- and post-synaptic rescue had even greater values for branch length and bouton number than WT controls. One possibility for this result is the fact that Myd is overexpressed by multiple Gal4 drivers in this condition. Together, these results

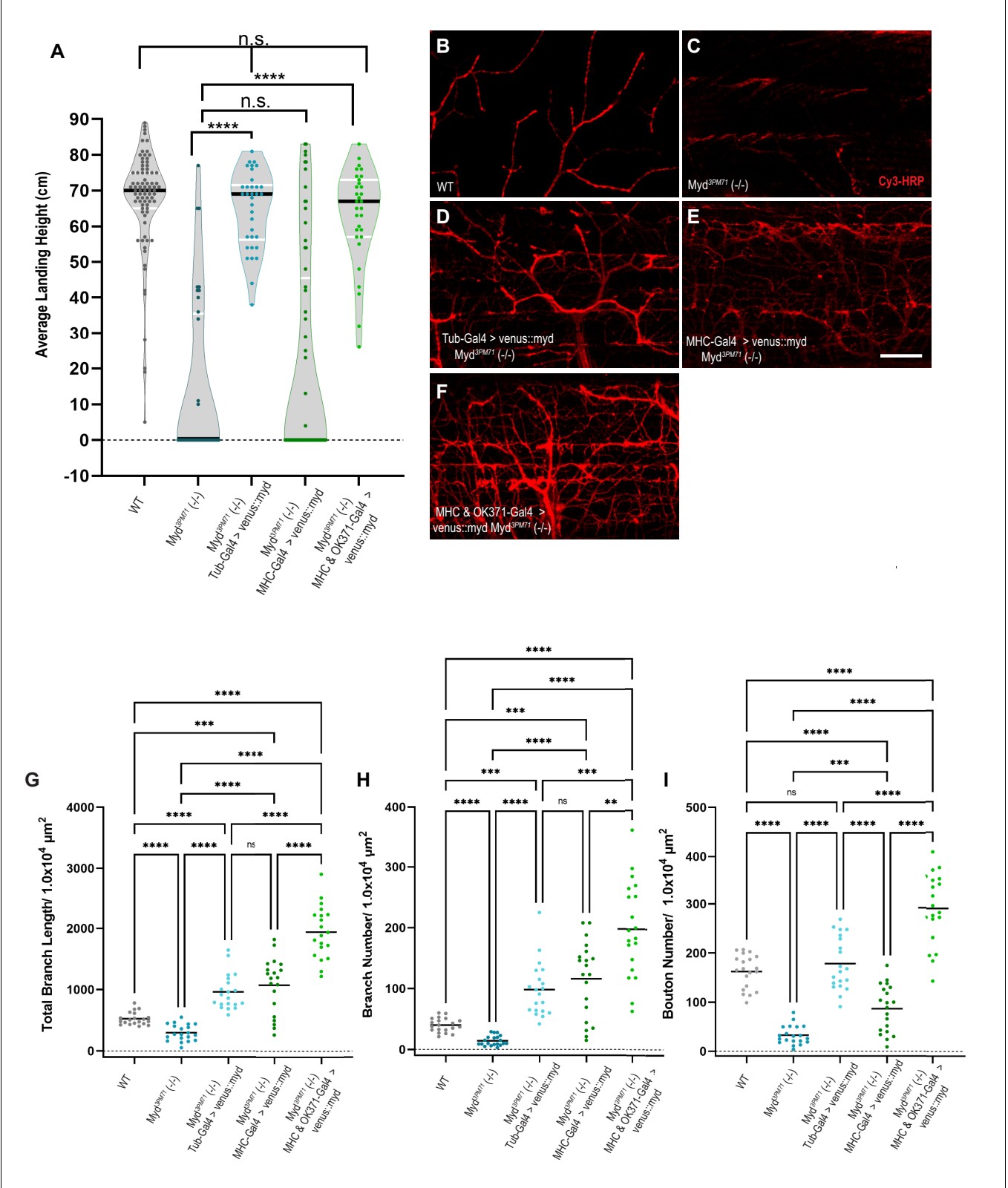

**Figure 5.** Mayday expression in muscles and motor neurons is necessary for synaptic integrity. (**A**) Average landing height of flies expressing UAS-venus::Myd either ubiquitously, in muscles, or in both muscles and motor neurons at Day 25 the in a *myd*^3PM71^ mutant background. The average landing height (cm) and the sample size (n) for each are as follows: WT 67.3 cm, n = 81, *myd*^3PM71^ 14.1 cm, n = 36, Tub-Gal4>venus::myd, *myd*^3PM71^ 64.8 cm, n = 34, MHC-Gal4>venus::myd, *myd*^3PM71^ 20.8 cm, n = 65, and MHC-Gal4 and OK371-Gal-4>venus::myd, *myd*^3PM71^ 63.2 cm and n = 31. (**B-F**)

*Figure 5 continued on next page*

*Figure 5 continued*
Confocal images of Myd rescue of gross morphology of DLM NMJs stained with Cy3-conjugated-HRP (red) at 63X. (**G-I**) Measurements of Total branch length, branch number, and bouton number for each condition, sample size n = 20 per genotype. The mean Total Branch Length (μM), mean branch number (n), and mean bouton number (n) for each genotype are as follows: WT 525.0 μM, n = 40, n = 163, *myd*[3PM71] 296.3 μM, n = 14, n = 33, Tub-Gal4>venus::myd, *myd*[3PM71] 962.9 μM, n = 98, n = 179, MHC-Gal4>venus::myd, *myd*[3PM71] 1082.0 μM, n = 116, n = 87, and MHC-Gal4 and OK371-Gal-4>venus::myd, *myd*[3PM71] 1952.0 μM, n = 198, n = 292, respectively. Black bars in A represent median values. Black bars in G-I represent mean values. ****, p-value<0.0001, ***, p-value<0.001, using a Brown-Forsythe and Welch ANOVA tests with Post hoc Dunnett's multiple comparisons. N.S. = Not Significant. Scale bar in E represents 20 μM for all images.
The online version of this article includes the following source data for figure 5:

**Source data 1.** Raw data values for *Figure 5*.

provide further evidence that Myd expression in muscles and motor neurons is necessary to maintain synaptic integrity over time.

## Mayday is functionally homologous to human Cab45

To determine whether Mayday is conserved across species, we performed a database search to align the protein sequence of Mayday. We found that Myd is very similar to the human protein Calcium Binding Protein of 45 kDa (Cab45), also known as Stromal Cell Derived Factor 4 (SDF4) (*Koivu et al., 1997*; *Scherer et al., 1996*; *von Blume et al., 2012*). Protein alignment comparing Myd with Cab45 revealed a 48% similarity and a 28% identity using the DRSC Integrative Ortholog Prediction Tool (DIOPT) (*Hu et al., 2011*). The regions of greatest similarity lie within the conserved EF hand

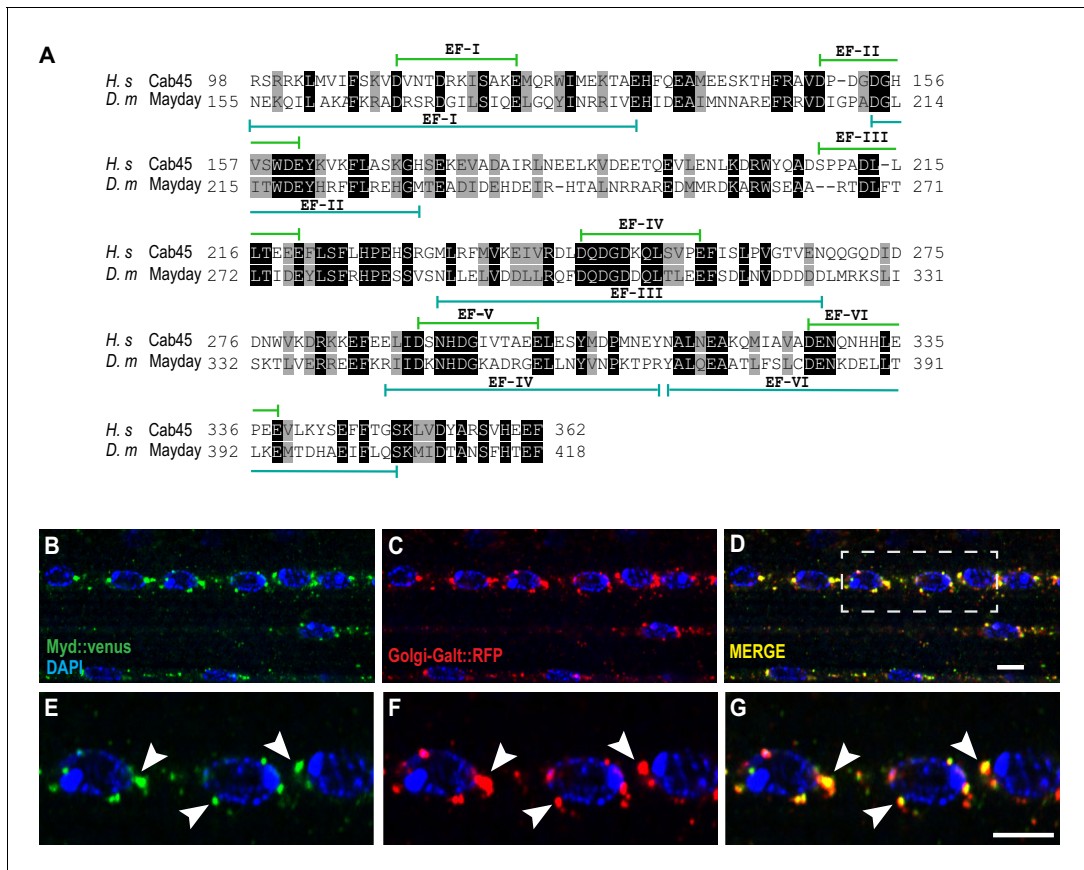

**Figure 6.** Mayday shares functional homology with human Cab45. (**A**) Protein alignment between human Cab45 (NP_057260.2) and *Drosophila* Myd (NP_001262725). Identical amino acids are represented by gray boxes and similarities are shaded black. (**B–G**) Confocal images of UAS-venus::Myd (green) co-localizing with UAS-Golgi-GalT-RFP (red) and DAPI (blue) at ×63 magnification. The white box in D represents the zoomed in region for panels E-G. Areas of colocalization are marked with white arrowheads. Scale bar in D = 10 μM for panels B-D. Scale bar in G = 6 μM for panels E-G.

domains (*Figure 6A*), which are associated with Ca$^{2+}$ binding (*Honoré, 2009*; *Honoré and Vorum, 2000*). Cab45 is ubiquitously expressed and localizes to the Trans Golgi Network (TGN), where it plays a role in sorting secretory cargo (*Crevenna et al., 2016*; *Koivu et al., 1997*; *Scherer et al., 1996*; *von Blume et al., 2012*). To determine if Myd has a similar function to Cab45, we first examined the cellular localization of Myd to see if it also localizes to the Golgi. We expressed UAS-Venus::myd along with a TGN-specific marker (UAS-Golgi-GalT::RFP) (*Zhou et al., 2014*) in muscle tissue and found that Myd strongly colocalizes with the Golgi marker (*Figure 6B–G*). This result suggests that Myd may function in a similar manner to Cab45.

Since Cab45 and Myd similarly localize to the TGN, we next investigated whether Myd might be functionally homologous to Cab45. Previous studies characterizing Cab45 have been primarily carried out using in vitro models that have been instrumental in elucidating the nature and role of Cab45 (*Crevenna et al., 2016*; *Koivu et al., 1997*; *Lam et al., 2007*; *Luo et al., 2016*; *Scherer et al., 1996*; *von Blume et al., 2012*). To test for functional homology between Myd and Cab45, we assessed whether expression of human Cab45 could rescue the defective flight phenotype in *myd$^{3PM71}$* mutants. To test this hypothesis, we generated and expressed UAS-Cab45 ubiquitously using Tub-Gal4 in a *myd$^{3PM71}$* mutant background. Similar to our results with Myd, ubiquitous expression of human Cab45 was able to rescue the flight defect in *myd$^{3PM71}$* mutants (*Figure 7A*), suggesting functional homology between Myd and Cab45.

We also tested whether expression of Cab45 specifically in muscles can rescue the flight phenotype seen in the *myd$^{3PM71}$* mutant background. Expressing Cab45 in muscle tissue alone was not able to restore flight ability (*Figure 7A*). Similar to our previous experiments with Myd expression, these results demonstrate that muscle-specific expression of Cab45 is insufficient to maintain synaptic integrity.

Since simultaneous expression of Myd in both muscles and motor neurons was able to restore the flight defect of *myd$^{3PM71}$* mutants, we tested whether expression of Cab45 in these tissues would provide a similar rescue. We found that expression of UAS-Cab45 in both muscles and motor neurons was able to restore flight when driven in a *myd$^{3PM71}$* mutant background (*Figure 7A*).

We next investigated whether expression of human Cab45 could also prevent the gross morphological deficits at NMJs in *myd$^{3PM71}$* mutants. Ubiquitous expression of UAS-Cab45 was able to restore the structural defects found in *myd* mutants (*Figure 7B–D*), as well as total branch length, branch number, and bouton number (*Figure 7G–I*). When we expressed Cab45 in muscle tissue alone in the homozygous mutant background (*Figure 7E*), it was unable to completely restore intact WT NMJs. Although total branch length and branch number were restored, the NMJs looked highly unorganized and lacked a restoration of boutons (*Figure 7G–I*). However, expression of Cab45 in both pre-and postsynaptic tissue was able to restore these values (*Figure 7F–I*). Together, this evidence supports the functional homology between Cab45 and Myd.

## Dysregulation of trans-synaptic BMP signaling in *myd* mutants

Given the requirement of Myd in muscle tissue and the loss of DLM innervation in *myd$^{3PM71}$* mutants, we next investigated how Myd may be involved in maintaining synaptic integrity. The localization of Myd to the TGN and its shared functional homology with Cab45 suggest that Myd could play a role in secreting a trans-synaptic signal between the postsynaptic muscle tissue and the presynaptic motor neuron. There are several well-studied signaling pathways involved in trans-synaptic communication, including Wingless, the *Drosophila* Wnt ortholog, and Glass bottom boat (Gbb) (*McCabe et al., 2003*; *Packard et al., 2002*). Because Myd has a clear role in muscles (*Figure 4A*), we specifically focused on retrograde signaling mechanisms. The *Drosophila* retrograde BMP signaling cascade begins with secretion of the morphogen Gbb from muscle tissue to the presynaptic motor neuron and regulates synaptic growth and development (*Aberle et al., 2002*; *Marqués et al., 2002*; *McCabe et al., 2004*; *McCabe et al., 2003*; *O'Connor-Giles et al., 2008*; *Rawson et al., 2003*; *Sweeney and Davis, 2002*). Gbb then binds to the BMP receptor, wishful thinking (wit), a BMP-Type II receptor, a transmembrane receptor serine-threonine kinase that phosphorylates BMP-Type I receptors Thickveins (Tkv) and/or saxophone (sax) at the presynaptic terminal and form a complex (*Aberle et al., 2002*; *Marqués et al., 2002*; *McCabe et al., 2003*; *Rawson et al., 2003*; *Sweeney and Davis, 2002*). Once the complex enters the presynaptic terminal, Tkv phosphorylates the R-Smad Mothers against Decapentaplegic (Mad) and forms a complex with Medea (*med*) and translocates to the nucleus to regulate transcription of target genes (*Chen et al., 1998*; *Das et al.,*

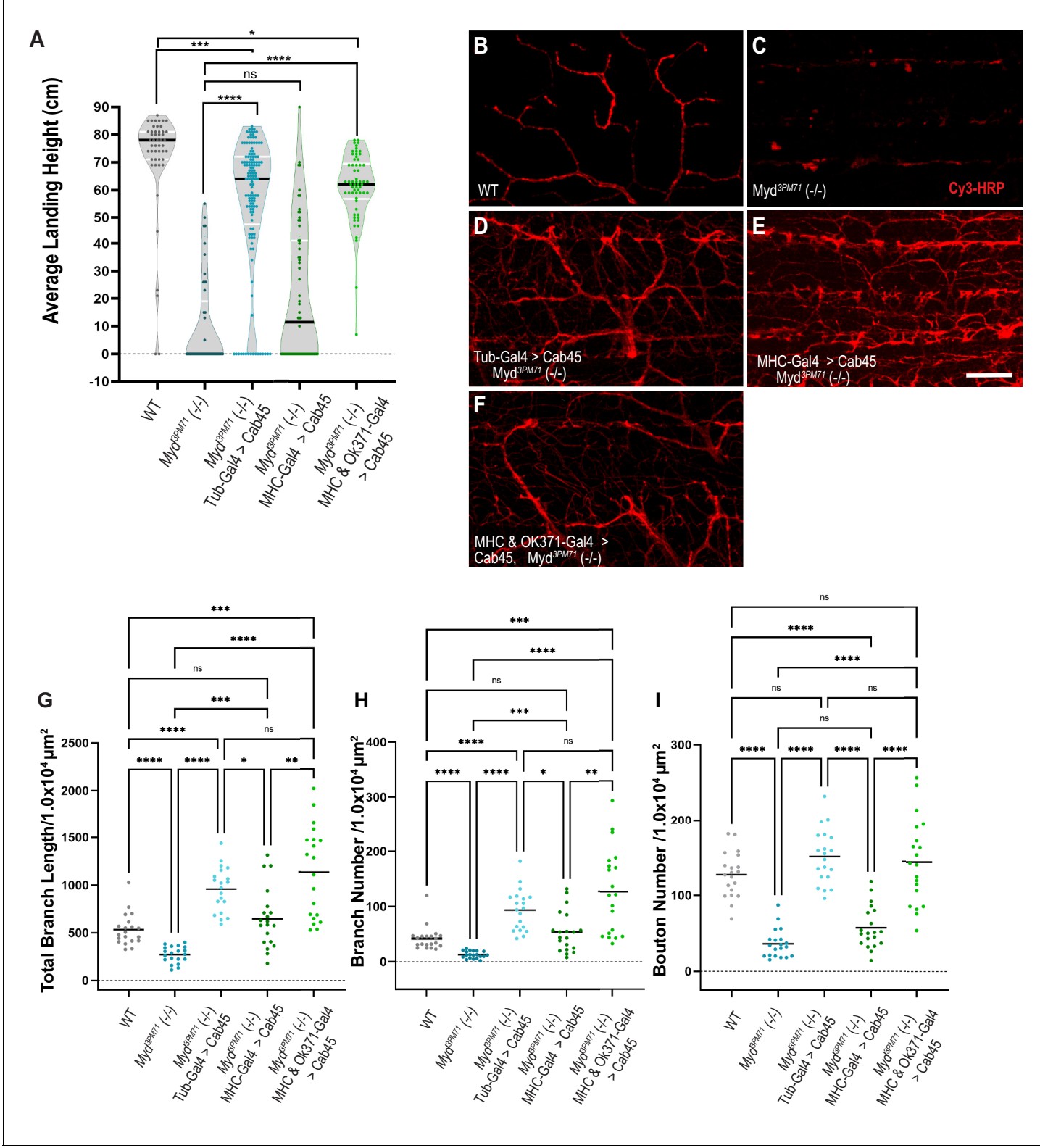

**Figure 7.** Human Cab45 expression rescues *myd* mutant phenotypes. (**A**) Average landing height of Day 25 Cab45 rescue flies. The average landing height (cm) and sample size (n) for each group are as follows: WT 71.4 cm, n = 49, *myd*$^{3PM71}$ 10.1 cm, n = 49, Tub-Gal4>Cab45, *myd*$^{3PM71}$ 55.6 cm, n = 137, MHC-Gal4>Cab45, *myd*$^{3PM71}$ 21.0 cm, n = 64, and MHC-Gal4 and OK371-Gal-4>Cab45, *myd*$^{3PM71}$ 60.1 cm, n = 62, respectively. (**B-F**) Confocal images of Cab45 expressed ubiquitously, in muscles, or in both muscles and motor neurons together in a *myd*$^{3PM71}$ mutant background. Neuronal

*Figure 7 continued on next page*

*Figure 7 continued*

membranes are labeled with Cy3-conjugated-HRP (red). (G-I) Measurements of Total branch length, branch number, and bouton number for each condition with a sample size of n = 20 for each genotype. For each group the mean Total branch length (μM), mean branch number (n), and mean bouton number (n), are listed as follows: WT 533.0 μM, n = 42, n = 128, myd³ᴾᴹ⁷¹271.3 μM, n = 13, n = 36, Tub-Gal4>Cab45, *myd³ᴾᴹ⁷¹* 957.9 μM, n = 94, n = 151, MHC-Gal4>Cab45, *myd³ᴾᴹ⁷¹*, 648.1 μM, n = 54, n = 57, and MHC-Gal4 and OK371-Gal-4>Cab45, *myd³ᴾᴹ⁷¹* 1138.0 μM, n = 128, n = 145. Black bars in **A** represent median values. Black bars in **G-I** represent mean values. ****, p-value<0.0001, ***, p-value<0.001, **, p-value<0.01, *, p-value<0.05 using Brown-Forsythe and Welch ANOVA tests with Post hoc Dunnett's multiple comparisons N.S. = Not Significant. Scale bar in **E** = 20 μM for panels **B-F**.

The online version of this article includes the following source data for figure 7:

**Source data 1.** Raw data values for *Figure 7*.

*1998*; *Hoodless et al., 1996*; *Inoue et al., 1998*; *Raftery et al., 1995*; *Wiersdorff et al., 1996*; *Wisotzkey et al., 1998*). However, it is unknown whether Gbb signaling is needed to maintain synaptic integrity in adults.

To determine if Gbb signaling is impaired at adult NMJs in *myd³ᴾᴹ⁷¹* mutants, we compared Gbb staining within DLMs in WT flies and *myd³ᴾᴹ⁷¹* mutants. At Day 3, we found Gbb to be present at both WT and myd NMJs, with the level of Gbb only slightly elevated in *myd³ᴾᴹ⁷¹* mutants (*Figure 8A–B*). By Day 25, the Gbb signal in WT flies is consistent with that seen at Day 3. However, in *myd³ᴾᴹ⁷¹* mutants, the Gbb signal at Day 25 is significantly enhanced compared to controls (*Figure 8C–E*), suggesting an accumulation of Gbb in the muscle tissue and potential dysregulation of BMP signaling.

To further investigate the regulation of BMP signaling in *myd³ᴾᴹ⁷¹* mutants, we tested for a dominant genetic interaction between *myd³ᴾᴹ⁷¹* and *gbb* using double heterozygous mutants. At Day 25, *myd³ᴾᴹ⁷¹*/+ and *gbb^D20*/+ (*Chen et al., 1998*) heterozygotes alone had a strong flight performance, while *myd³ᴾᴹ⁷¹* and *gbb^D20* double heterozygous mutants exhibited a severe flight phenotype (*Figure 8F*). The flight performance of double heterozygous mutants provides evidence of a genetic interaction between *gbb* and *myd,* suggesting that the role of *myd* in maintaining synaptic integrity involves trans-synaptic BMP signaling.

To determine where the increased Gbb is accumulating within DLMs, we measured co-localization of the Gbb puncta with various membrane-bound cellular structures including golgi, endosomes, and lysosomes. We identified a small but significant increase in the amount of Gbb located within either golgi or early endosomes at Day 25 in *myd³ᴾᴹ⁷¹* mutants (*Figure 8G–M*). However, we did not find any significant accumulation with lysosomes. The accumulation of Gbb within muscle golgi and early endosomes is consistent with proteins that are sorted and trafficked. Interestingly, the vast majority of Gbb puncta did not co-localize with any of the examined membrane-bound structures, suggesting that much of this Gbb is likely located within the cytoplasm.

## Mayday genetically interacts with presynaptic components of the BMP signaling cascade

We further investigated the role of *myd* in trans-synaptic BMP signaling by measuring the readout of downstream BMP targets. The most commonly used readout of BMP signaling activity across synapses is the staining of phosphorylated mothers against DPP (pMad) (*McCabe et al., 2003*). If Gbb accumulates in postsynaptic muscle tissue due to a defect in trans-synaptic signaling, we hypothesized that this would then result in a decrease in pMad activity in presynaptic motor neuron nuclei. We examined pMad staining at an early time point and found similar staining between WT and *myd³ᴾᴹ⁷¹* flies (*Figure 9A–F*), suggesting that BMP signaling is not impaired in *myd³ᴾᴹ⁷¹* mutants at early time points. However, at Day 21 *myd³ᴾᴹ⁷¹* mutants had a significant decrease in pMad signaling in motor neuron nuclei compared to WT (*Figure 9G–M*). The decrease in nuclear pMad suggests that trans-synaptic BMP signaling is progressively impaired in *myd* mutants. This result, supported by the accumulation of Gbb in post-synaptic muscle tissue (*Figure 8C*), suggest that Gbb cannot reach the presynaptic motor neuron to activate the downstream signaling components.

To distinguish between nuclear and synaptic pMAD, we also measured pMAD signaling located at NMJs. At Day 3, we found small amounts of pMad localized primarily within postsynaptic muscle tissue (*Figure 9N–O*). However, at Day 21, there was a significant increase in pMAD staining within

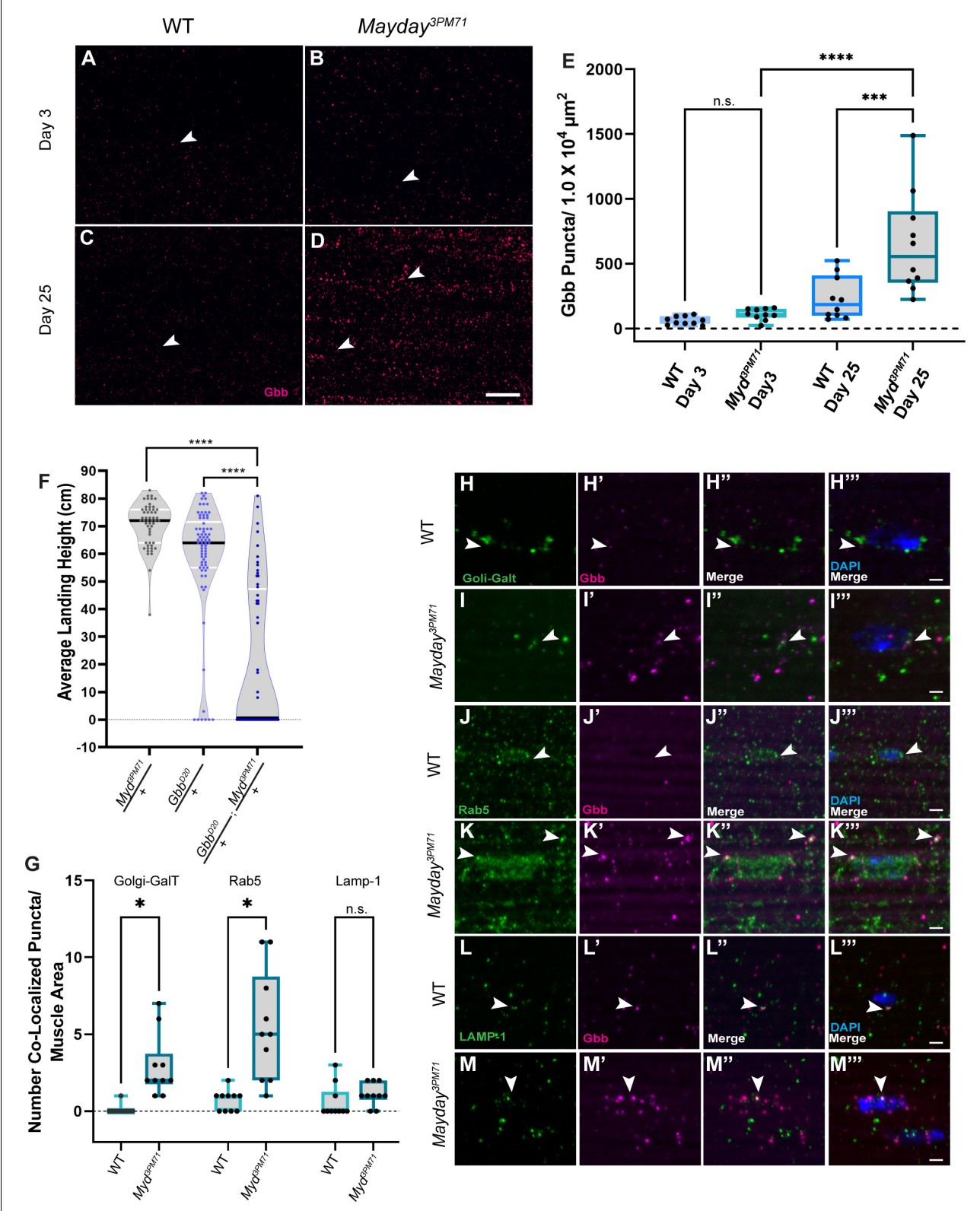

**Figure 8.** *Myd* genetically interacts with *gbb*. (A–D) Confocal images of DLM NMJs stained with Gbb (Magenta) at early (A–B) and late (C–D) time points at 63X. Arrowheads designate Gbb puncta in each image. (E) Quantification of the number of Gbb puncta per muscle area for each genotype, sample size n = 10. Box plots show the distribution of data with bar from the min to the max with the mean Gbb puncta for each condition as follows: WT Day 3 n = 61, *myd*$^{3PM71}$ Day 3 n = 110, WT day 25 n = 234, and *myd*$^{3PM71}$ Day 3 n = 652. (F) Progressive loss of flight at day 25 in heterozygous and

*Figure 8 continued on next page*

*Figure 8 continued*

double heterozygous mutants for $myd^{3PM71}$ and $gbb^{D20}$. The average landing height (cm) and sample size (n) for each condition: $myd^{3PM71}$ /+ 70.5 cm, n = 54, $gbb^{D20}$/+ 58.3 cm, n = 74, and $gbb^{D20}$/+; $myd^{3PM71}$ /+ 21.1 cm, n = 56. (G–M) Quantification of Gbb puncta located within Golgi (H–I), early endosomes (J–K), or lysosomes (L–M) with n = 10 per condition at Day 25. Box plots show the distribution of data with a bar from the min to the max, with the mean colocalized Gbb puncta for each comparison, Golgi: WT 0.01, $myd^{3PM71}$ 2.9, early endosomes WT 0.7, $myd^{3PM71}$ 5.5, and lysosomes WT 0.6, and $myd^{3PM71}$ 1.1, respectively. Arrowheads designate Gbb puncta co-localization. In **E** ****, <0.0001 p-value, ***<0.001 p-value using a one-way ANOVA with Turkey Post hoc multiple comparisons. N.S. = Not Significant. In **F** black bars in graphs represent median values. In **F** and **G** ****, p-value<0.0001, *, p-value<0.05 using Brown-Forsythe and Welch ANOVA tests with Post hoc Dunnett's multiple comparisons. N.S. = Not Significant Scale bar in **D** represents 20 µM for panels **A-D**. Scale bar in **M** represents 2 µM for **H-M**.

The online version of this article includes the following source data for figure 8:

**Source data 1.** Raw data values for *Figure 8*.

muscles in $myd^{3PM71}$ mutants (*Figure 9P–R*). Thus, the decrease in presynaptic nuclear pMAD signaling is accompanied by an increase in postsynaptic signaling.

To further characterize the relationship between myd and downstream BMP signaling components, we tested for dominant genetic interactions between these genes by assessing flight performance. We first looked at the potential genetic interaction with *wishful thinking (wit)*, the first target of the BMP signaling cascade (*Aberle et al., 2002*; *Marqués et al., 2002*). At day 28, flies heterozygous for *wit* ($wit^{B11}$) flew well. By contrast, double heterozygous combinations of $myd^{3PM71}$ and $wit^{B11}$ showed a poor flight performance (*Figure 9S*). We next investigated if *myd* also interacts with *thickveins (tkv)*, which encodes a Type 1 BMP receptor that forms a complex with Wit (*Aberle et al., 2002*; *Allan et al., 2003*; *Baines, 2004*; *McCabe et al., 2003*; *Ruberte et al., 1995*; *Sweeney and Davis, 2002*). Similar to our results with *wit*, we also found that double heterozygous combinations of $myd^{3PM71}$ and two independent alleles of *tkv* ($tkv^7$ and $tkv^8$) had a progressive loss of flight while heterozygotes alone performed well (*Figure 9T*). We continued to test the potential genetic interaction between *myd* and *mad*, the effector that is phosphorylated by *tkv* (*Inoue et al., 1998*; *McCabe et al., 2003*; *Wiersdorff et al., 1996*). The $mad^{-1-2}$ (*Wiersdorff et al., 1996*) and $myd^{3PM71}$ heterozygotes flew well alone, while the $mad^{-1-2}$ and $myd^{3PM71}$ double heterozygotes had a flight deficit (*Figure 9U*). These genetic interactions further demonstrate the role of myd in the BMP signaling cascade.

If the underlying defect in myd mutants is the inability to secret Gbb from muscle tissue in a retrograde fashion, we would predict that expressing a constitutively active form of the BMP receptor Tkv (*Hoodless et al., 1996*) in presynaptic motor neurons could restore synaptic integrity. To test this, we expressed constitutively active Tkv ($UAS$-$Tkv^{CA}$) in motor neurons using the $OK371$-$Gal4$ driver (*Mahr and Aberle, 2006*) in a $myd^{3PM71}$ mutant background, and found that flight behavior was restored even at Day 28 (*Figure 9V*). The ability of presynaptic BMP signaling to rescue the flight defect in $myd^{3PM71}$ mutants supports the functional role of *myd* in trans-synaptic BMP signaling to maintain synaptic integrity in adult NMJs.

## Loss of retrograde BMP signaling in *mayday* mutants results in cell death

Our model that *myd* positively regulates retrograde BMP signaling infers that Myd has a neuroprotective role in maintaining synaptic integrity. Since previous studies with Cab45 have demonstrated a role in preventing apoptosis (*Chen et al., 2014*; *Grønborg et al., 2006*; *Shen et al., 2018*), we wanted to determine if cell death is a consequence of mutations in *mayday*. To further evaluate the consequences of impaired BMP signaling in $myd^{3PM71}$ mutants, we assessed the viability of both muscle cells and motor neurons using TUNEL staining to detect cell death. While we found a very small number of TUNEL-positive cells within motor neuron nuclei at Day 25, there was widespread TUNEL-positive nuclei in $myd^{3PM71}$ samples (*Figure 10A–G*). We also detected muscle cell death as assessed by the presence of TUNEL-positive cells in DLMs of $myd^{3PM71}$ mutants in comparison to WT flies (*Figure 10H–N*). These results suggest that the defects in synaptic integrity found in $myd^{3PM71}$ mutants leads to the loss of both presynaptic motor neurons as well as postsynaptic muscle tissue.

Thus, these results support the model that Mayday sustains trans-synaptic signaling in adult NMJs. According to our model, we propose that Myd regulates the trafficking of Gbb from the

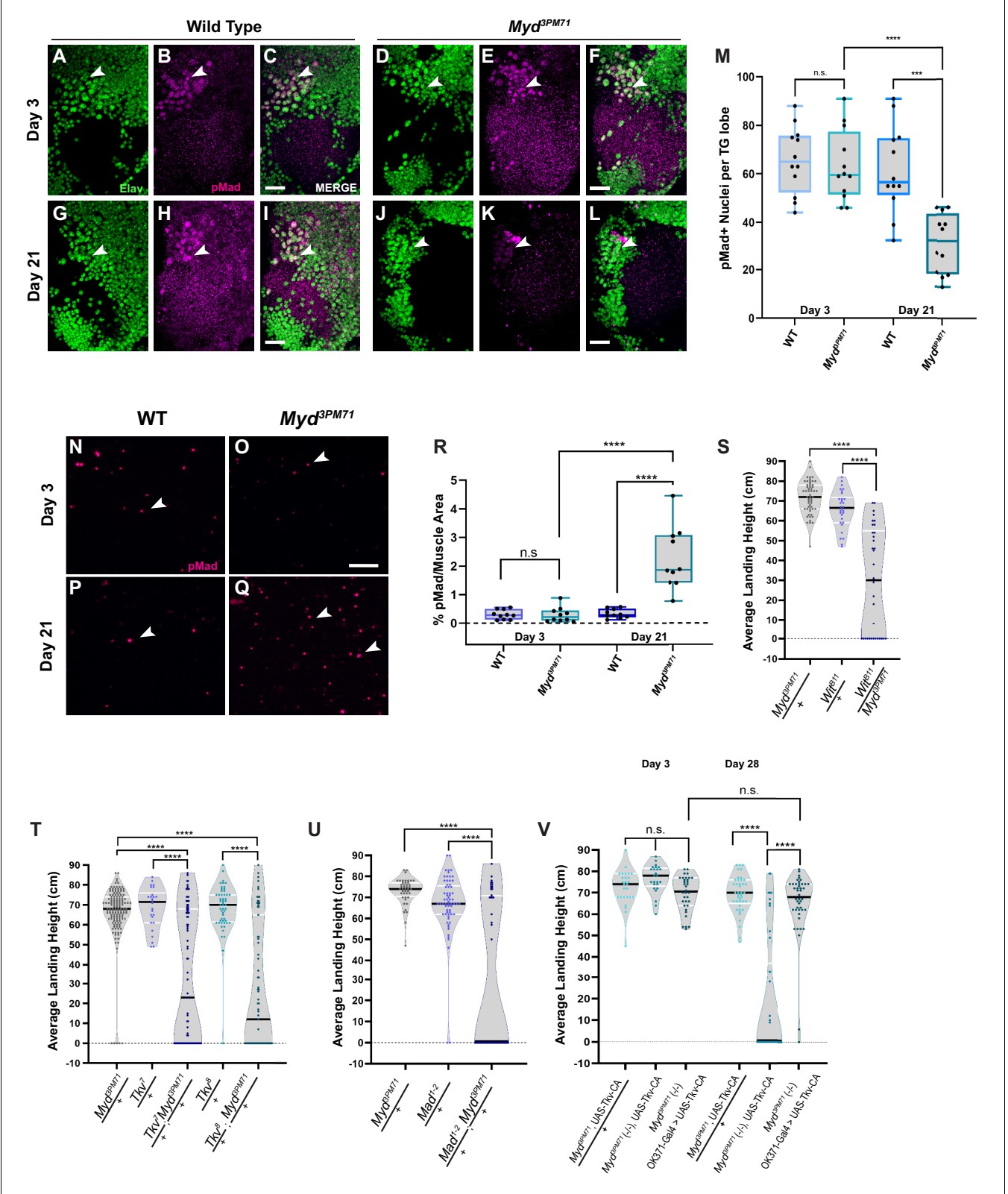

**Figure 9.** *Myd* is a positive regulator of trans-synaptic BMP signaling. (A–L) Confocal images of motor neuron nuclei in *myd³ᴾᴹ⁷¹* and controls stained with elav (green) and pMad (magenta) at ×40 magnification. Arrowheads designate areas of co-localization. (M) Quantification of pMad-positive motor neuron nuclei within each lobe of the thoracic ganglion with a sample size of n = 12 per condition. The mean number of pMad+ nuclei per condition: WT Day 3 n = 66, *myd³ᴾᴹ⁷¹* Day 3 n = 63, WT Day 21 n = 62, and *myd³ᴾᴹ⁷¹* Day 21 n = 31. (N–Q) Images of DLM NMJs stained with pMad (magenta) at

Figure 9 continued

×63 magnification. Arrowheads designate pMad puncta. (R) Quantification of the amount of pMad signaling per muscle area (2916 μM$^2$), n = 10 for each condition. The mean percent pMad/Muscle Area per condition: WT Day 3 = 0.3%, myd$^{3PM71}$ Day 3 = 0.3%, WT Day 21 = 0.3% and myd$^{3PM71}$ Day 21 = 2.3%. (S) Progressive loss of flight in wit and myd double heterozygous combinations at Day 28. (T) Progressive loss of flight in tkv and myd double heterozygous combinations at Day 28. (U) Progressive loss of flight in mad and myd double heterozygous combinations at Day 28. (V) Motor neuron expression of a constitutively active Tkv in myd$^{3PM71}$ mutants. Average landing height (cm) and sample size (n) can be found in *Supplementary file 2* for panels (S–T). Black bars in graphs represent median values. For panels **M** and **R**. ****, p-value<0.0001, ***, p-value<0.001,. using a one-way ANOVA with Turkey Post hoc multiple comparisons. N.S. = Not Significant. For **S-T**, ****, p-value<0.0001 using Brown-Forsythe and Welch ANOVA tests with Post hoc Dunnett's multiple comparisons. N.S. = Not Significant Scale bar in **L** represents 20 μM for A-L. Scale bar in **O** represents 10 μM for panels N-Q.

The online version of this article includes the following source data for figure 9:

**Source data 1.** Raw data values for *Figure 9*.

TGN. Myd promotes the secretion of Gbb from the muscle tissue to the presynaptic motor neurons. Once Gbb binds to Wit, Wit then phosphorylates Tkv. The receptor complex enters the presynaptic terminal and Tkv phosphorylates Mad. Mad then forms a complex with Medea and translocated to the nucleus to regulate transcription of BMP signaling. By contrast, in myd$^{3PM71}$ mutants BMP retrograde signaling is dysregulated by the accumulation of Gbb in the muscle tissue. As a result, Gbb accumulates in the postsynaptic muscle tissue. Because Gbb is not adequately transferred to presynaptic motor neurons, presynaptic pMAD levels decrease. Over time, the excess Gbb in muscles and the decrease in pMAD levels in motor neurons lead the loss of synaptic integrity, leading to the death of both muscle tissue and motor neurons (*Figure 11A–B*). These results are consistent with previous studies linking an imbalance of BMP signaling to neuromuscular dysfunction. For example, elevated BMP signaling was found in individuals with muscular dystrophy (2017; *Bernasconi et al., 1995*; *Yamazaki et al., 1994*). Additionally, enhanced pMAD signaling within muscle tissue has also been used as a biomarker for Amyotrophic Lateral Sclerosis (ALS) (*Si et al., 2014*; *Si et al., 2015*). Thus, our model combined with these lines of evidence supports the idea that myd has a neuroprotective role in positively regulating BMP signaling in postsynaptic muscle tissue in adult NMJs.

## Discussion

Our current study describes *mayday* (*myd*), a previously uncharacterized gene that plays a role in maintaining synaptic integrity with age by promoting trans-synaptic signaling. We found that myd$^{3PM71}$ mutants have structural and functional deficits in adult DLM NMJs. Through tissue-specific RNAi and rescue experiments, we determined that Myd is necessary in both postsynaptic muscle tissue and presynaptic motor neurons to maintain synaptic integrity. Myd localizes to the TGN and shares functional homology with human Cab45. We found that Myd sustains retrograde BMP signaling in adult DLM NMJs through genetic interactions with *gbb*, *tkv*, *wit*, and *mad* mutants and staining of Gbb and pMad markers. Finally, myd sustains the viability of presynaptic motor neurons and postsynaptic muscles.

From developmental studies, we have learned that Gbb is a morphogen secreted in a retrograde manner trans-synapticaly from postsynaptic muscle tissue to the presynaptic motor neuron in larval NMJs to promote synaptic growth (*Aberle et al., 2002*; *Marqués et al., 2002*; *McCabe et al., 2004*; *McCabe et al., 2003*; *Rawson et al., 2003*; *Sweeney and Davis, 2002*). However, relatively little is known regarding the roles of this pathway in fully developed organisms. Recent evidence demonstrates that sustained BMP signaling is required to maintain FMRFamide expression in a subset of neurons in the *Drosophila* brain (*Eade and Allan, 2009*). Our results here further demonstrate that retrograde BMP signaling that regulates NMJ development is required in adult NMJs to sustain synaptic integrity with age. It is also possible that several other signaling pathways crucial for organism development may be required throughout the life of the organism.

Our knockdown and rescue experiments using *myd* demonstrate that it maintains synaptic integrity through roles in both pre- and postsynaptic tissue. While most studies involving BMP in synaptic growth report a retrograde signaling mechanism, recent evidence suggests that this pathway could also signal in an anterograde fashion (*Dudu et al., 2006*; *Goold and Davis, 2007*). While our genetic studies provide support for retrograde BMP signaling, we cannot rule out the possibility that

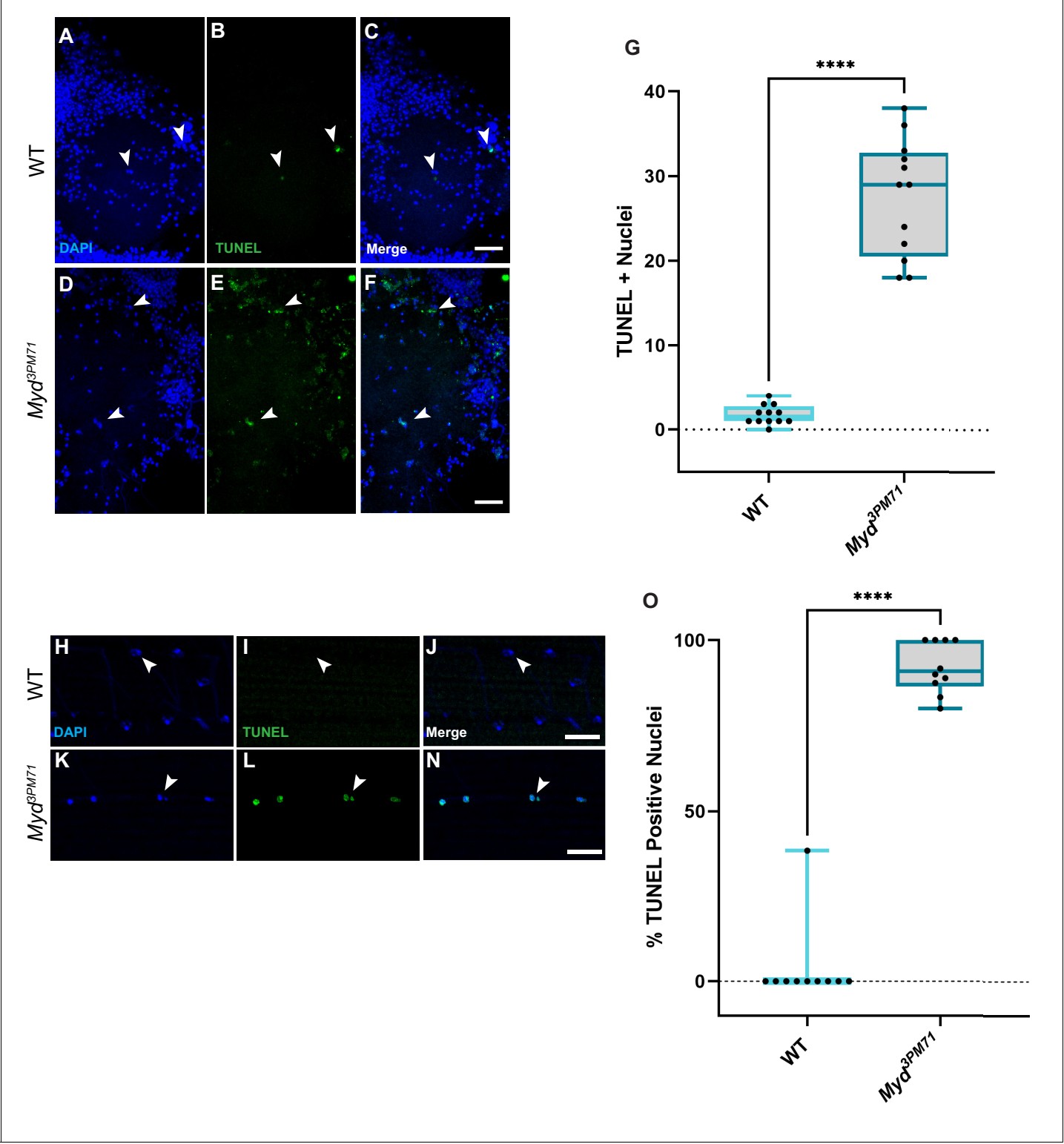

**Figure 10.** Myd sustains motor neuron and muscle viability. (A–F) Confocal images of Day 25 adult thoracic ganglia stained with DAPI (blue) and TUNEL (green) at ×40. White arrows highlight areas of cell death. (G) Quantification of the number of TUNEL-positive nuclei located within a single lobe of the thoracic ganglion with a sample size n = 12, in each condition. The box plot shows the distribution of data with the min and max, with the mean number of TUNEL + nuclei for WT n = 2 and myd*3PM71* n = 28. (H–N) Confocal images of Day 28 adult DLMs stained with DAPI (blue) and TUNEL (green) at ×63. White arrows highlight areas of cell death. (O) Quantification of the percentage of TUNEL-positive nuclei within each muscle cell. The box plot shows the distribution of data with the min and max, with the mean % of number of TUNEL+ nuclei for WT 3.8% and myd*3PM71* 92.0%,

*Figure 10 continued on next page*

*Figure 10 continued*

respectively. Scale bar in **F** represents 20 µM for panels **A-F**, and the scale bar in **N** represents 10 µM for panels H-N. ****p<0.0001 using a Student's T-Test.

The online version of this article includes the following source data for figure 10:

**Source data 1.** Raw data values for *Figure 10*.

anterograde BMP signaling also plays an important role in maintaining synaptic integrity. In particular, the levels of pMAD that we observed were present within both presynaptic motor neuron terminals as well as postsynaptic muscles. Further studies aimed at characterizing BMP signal activation within muscle cells should help with our understanding of the mechanisms responsible for synaptic maintenance.

While trans-synaptic BMP signaling plays a clear role in maintaining synapses, *myd* mutations likely impair other pathways associated with cargo trafficking. In addition to secretory cargo, Cab45 also has a role in trafficking lysosomal proteases (*Crevenna et al., 2016*; *von Blume et al., 2011*; *von Blume et al., 2012*). Given the functional homology shared between Cab45 and Myd, it is possible that the trafficking of these lysosomal hydrolases needed for autophagy could be disrupted. Defects in autophagy have been strongly linked with neurodegenerative diseases (*Hara et al., 2006*; *Komatsu et al., 2006*). Therefore, it is possible that *myd* mutants have disruptions in autophagy that lead to the loss of synaptic integrity. It will be interesting to investigate how Myd impacts these other processes that are associated with neuronal dysfunction.

Our assessment of synaptic dysfunction in the current study includes flight performance as a readout of functional integrity, as well as morphological measurements of branch length, branch number, and bouton number using a presynaptic membrane marker. In future studies, it will be helpful to further evaluate synaptic integrity in *mayday* mutants. Additional functional assays may include electrophysiological measurements of activity across these NMJs, and more structural data could be obtained through the use of a wide array of synaptic markers, as well as ultrastructural analysis using Transmission Electron Microscopy. Together, these types of studies should allow for an even greater understanding of synaptic dysfunction and the mechanisms required to maintain these critical structures.

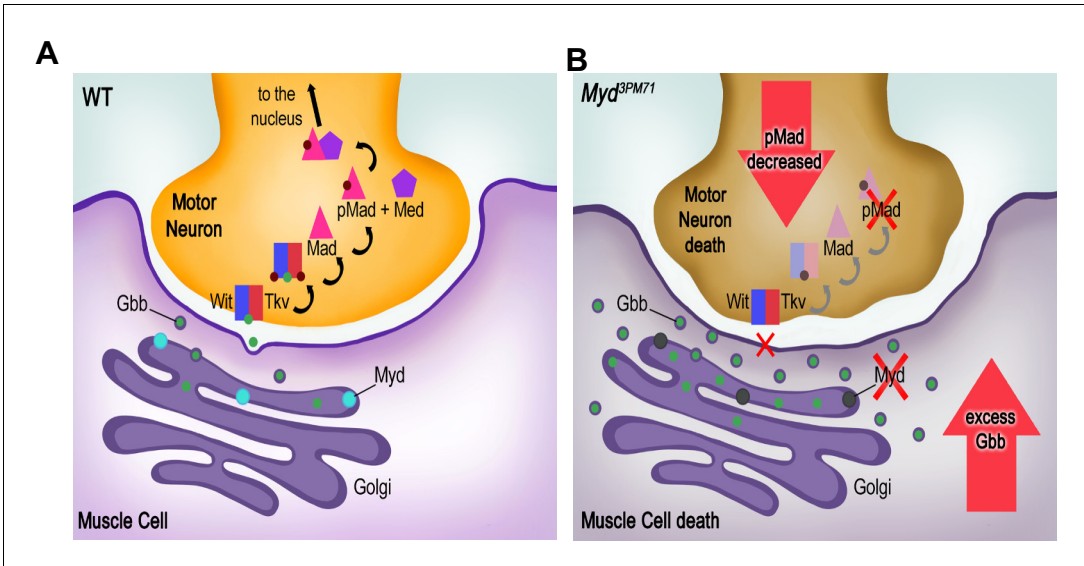

**Figure 11.** Summary of synaptic defects in *myd³ᴾᴹ⁷¹* mutants. (**A**) In WT flies, Myd is required for proper retrograde trafficking of Gbb. Successful trafficking of Gbb to presynaptic motor neurons and subsequent BMP signaling promotes synaptic maintenance. (**B**) In *myd³ᴾᴹ⁷¹* flies, defects in Gbb trafficking result in decreased BMP signaling in motor neurons along with accumulated Gbb in muscles, leading to cell death in motor neurons and muscles, respectively.

# Materials and methods

## Key resources table

| Reagent type (species) or resource | Designation | Source or reference | Identifiers | Additional information |
|---|---|---|---|---|
| Genetic reagent (*D. melanogaster*) | Oregon R | BDSC: 5 | | |
| Genetic reagent (*D. melanogaster*) | Df(3R)ED5938 | BDSC:24139 | w[1118]; Df(3R)ED5938, P{w[+mW.Scer\FRT.hs3]=3'.RS5+3.3'}ED5938/TM6C, cu[1] Sb[1] | |
| Genetic reagent (*D. melanogaster*) | CG31475[MI08258] | BDSC:51072 | y[1] w[*]; Mi{y[+mDint2]=MIC}CG31475[MI08258] | |
| Genetic reagent (*D. melanogaster*) | CG31475[MI08666] | BDSC:51100 | y[1] w[*]; Mi{y[+mDint2]=MIC}CG31475[MI08666] | |
| Genetic reagent (*D. melanogaster*) | CG31475[MB03509] | BDSC:24073 | w[1118]; Mi{ET1}CG31475[MB03509] | |
| Genetic reagent (*D. melanogaster*) | Tubulin-Gal4 | BDSC:5138 | y[1] w[*]; P{w[+mC]=tubP-GAL4}LL7/TM3, Sb[1] Ser[1] | |
| Genetic reagent (*D. melanogaster*) | Elav[C155]-Gal4 | BDSC:458 | P{w[+mW.hs]=GawB}elav[C155] | |
| Genetic reagent (*D. melanogaster*) | MHC-Gal4 | BDSC:55132 | P{w[+mC]=Mhc-GAL4.K}1, w[*]/FM7c | |
| Genetic reagent (*D. melanogaster*) | MHC-Gal4 | BDSC:55133 | w[*]; P{w[+mC]=Mhc-GAL4.K}2/TM3, Sb[1] | |
| Genetic reagent (*D. melanogaster*) | OK371-Gal4 | BDSC:26160 | OK371-Gal4 w[1118]; P{w[+mW.hs]=GawB}VGlut[OK371] | |
| Genetic reagent (*D. melanogaster*) | Repo-Gal4 | BDSC:7415 | Repo-Gal4 w[1118]; P{w[+m*]=GAL4}repo/TM3, Sb[1] | |
| Genetic reagent (*D. melanogaster*) | UAS-Dicer2 | BDSC:24651 | w[1118]; P{w[+mC]=UAS-Dcr-2.D}10 | |
| Genetic reagent (*D. melanogaster*) | UAS-Golgi-Galt-RFP | BDSC:65251 | w[1118]; P{w[+mC]=UAS-GalT-TagRFP-T}2; TM2/TM6B, Tb[1] | |
| Genetic reagent (*D. melanogaster*) | Gbb[D20] | BDSC:63054 | Gbb[D20] y[1] w[*]; P{w[+mW.hs]=FRT(w[hs])}G13 sha[1] gbb[D20]/SM6a | |
| Genetic reagent (*D. melanogaster*) | UAS-Tkv[CA] | BDSC:36537 | UAS-Tkv.[CA] w[*]; P{w[+mC]=UAS-tkv.CA}3 | |
| Genetic reagent (*D. melanogaster*) | Tkv[7] | BDSC:3242 | Tkv[7] tkv[7] cn[1] bw[1] sp[1]/CyO | |
| Genetic reagent (*D. melanogaster*) | Tkv[8] | BDSC:34509 | Tkv[8] tkv[8] cn[1] bw[1] sp[1]/CyO, P{ry[+t7.2]=sevRas1.V12}FK1 | |
| Genetic reagent (*D. melanogaster*) | Wit[B11] | BDSC:5174 | Wit[B11] bw[1]; wit[B11] st[1]/TM6B, Tb[1] | |
| Genetic reagent (*D. melanogaster*) | Mad[1-2] | BDSC:7323 | Mad[1-2] w[*]; Mad[1-2] P{ry[+t7.2]=neoFRT}40A/CyO | |
| Genetic reagent (*D. melanogaster*) | TsGal80 | BDSC:7108 | w[*]; P{w[+mC]=tubP-GAL80[ts]}10; TM2/TM6B, Tb[1] | |

*Continued on next page*

*Continued*

| Reagent type (species) or resource | Designation | Source or reference | Identifiers | Additional information |
|---|---|---|---|---|
| Genetic reagent (*D. melanogaster*) | TsGal80 | BDSC:7017 | w[*]; P{w[+mC]= tubP-GAL80[ts]}2/TM2 | |
| Genetic reagent (*D. melanogaster*) | UAS-luciferase | BDSC:35788 | y[1] v[1]; P{y[+t7.7] v[+t1.8]=UAS-LUC. VALIUM10}attP2 | |
| Genetic reagent (*D. melanogaster*) | UAS CG31475 $^{RNAi}$ | VDRC: 106664 | UAS-mayday$^{RNAi}$ | |
| Genetic reagent (*D. melanogaster*) | UAS-venus::myd | This Paper | | |
| Genetic reagent (*D. melanogaster*) | UAS-Cab45 | This Paper | | |
| Genetic reagent (*D. melanogaster*) | *Myd$^{3PM71}$* | This Paper | | |
| Cell line (*H. sapiens*) | HeLa | ATCC | Cat# CCL-2 | |
| Transfected construct (*E. coli*) | pBID-UASC-VG vector | PMID:22848718 | RRID:Addgene_35206 | |
| Transfected construct (*E. coli*) | pBID-UASC-G vector | PMID:22848718 | RRID:Addgene_35202 | |
| Antibody | Anti-DsRed (rabbit polyclonal) | Takara Bio USA Inc | Cat#: 632496 | (1:500), 24 hr, 48 hr with anti-Gbb |
| Antibody | Anti-GFP (chicken polyclonal) | ThermoFisher | Cat#: A10262 | (1:500) 24 hr |
| Antibody | Anti-Gbb (mouse monoclonal) | DSHB | Cat#: 3D6-24 | (1:500), 48 hr |
| Antibody | Anti-elav (rat monoclonal) | DSHB | Cat# Elav-9F8A9 | (1:20), 24 hr |
| Antibody | Anti-pMad (rabbit monoclonal) | Abcam | Cat#: ab529031 | (1:500), 24 hr |
| Antibody | Anti-lamp-1 (rabbit polyclonal) | Abcam | Cat#: ab30687 | (1:500), 48 hr |
| Antibody | Anti -rab5 (rabbit polyclonal) | Abcam | Cat#: ab31261 | (1:200), 48 hr |
| Antibody | Alexa Fluor 568 (goat anti rabbit IgG) | Invitrogen, Life Technologies | Cat#: A11036 | (1:200), 2 hr |
| Antibody | Alexa Fluor 488 (goat anti rabbit IgG) | Invitrogen, Life Technologies | Cat#: A11008 | (1:200), 2 hr |
| Antibody | Alexa Fluor 488 (goat anti chicken IgG) | Invitrogen, Life Technologies | Cat#: A11039 | (1:200), 2 hr |
| Antibody | Alexa Fluor 488 (goat anti mouse IgG) | Invitrogen, Life Technologies | Cat#: A11001 | (1:200), 2 hr |
| Antibody | Alexa Fluor 568 (goat anti mouse IgG) | Invitrogen, Life Technologies | Cat#: A11031 | (1:200), 2 hr |
| Antibody | Cy3-conjugated anti HRP | Jackson Laboratories | Cat#: 123-165-021 | (1:500), 2 hr |

*Continued on next page*

*Continued*

| Reagent type (species) or resource | Designation | Source or reference | Identifiers | Additional information |
|---|---|---|---|---|
| Antibody | FITC-conjugated anti HRP | Jackson Laboratories | Cat#: 123-545-021 | (1:200), 2 hr |
| Recombinant DNA | CG31475-Gold cDNA clone | *Drosophila* Genomics Resource Center | Cat#:16308 | |
| Sequence-based reagent | Cab45 Forward-5'-ATGGTCTGGC CCTGGGTG-3' | This Paper | IDT | Cab45 Isolation |
| Sequence-based reagent | Cab45 Reverse-5'-TCAAAACTCCT CGTGCACGCT-3' | This Paper | IDT | Cab45 Isolation |
| Sequence-based reagent | Actin5C Forward 5'- CGAAGAAGTTGCT GCTCTGGTTGT-3' | PMID:25823231 | IDT | qRT-PCR |
| Sequence-based reagent | Actin5C Reverse 5'- GGACGTCCCACAA TCGATGGGAAG-3' | PMID:25823231 | IDT | qRT-PCR |
| Sequence-based reagent | CG31475-Forward 5'- TCCAGGAATTGG GGCAGTACATAAATC-3' | PMID:17625558 | IDT | qRT-PCR |
| Sequence-based reagent | CG31475-Reverse 5' CTCGGGATGGC GGAAACTCA-3' | PMID:17625558 | IDT | qRT-PCR |
| Commercial assay or kit | In Situ Cell Death Detection Kit, Fluorescein | Millipore Sigma | Cat#: 11684795910 | |
| Commercial assay or kit | pCR8 Gateway cloning kit | ThermoFisher | Cat#:250020 | |
| Commercial assay or kit | Monarch Total RNA Isolation kit | New England Biolabs | Cat#: T2010 | |
| Commercial assay or kit | Monarch RNA Clean Up Kit | New England Biolabs | Cat:# T2030 | |
| Chemical compound, drug | VECTASHIELD Antifade Mounting Medium | Vector Laboratories | Cat#:H-1000 | |
| Chemical compound, drug | qScript cDNA synthesis SuperMix | Quantabio | Cat#: 95048–025 | |
| Chemical compound, drug | PowerUP SYBR Green Master Mix | Applied Biosystems | Cat#: A-25741 | |
| Chemical compound, drug | SuperScript III Reverse Transcriptase | Invitrogen, Life Technologies | Cat#: 18080093 | |
| Chemical compound, drug | Rnase OUT | Invitrogen, Life Technologies | Cat#: 10777019 | |
| Chemical compound, drug | Rnase H | New England Biolabs | Cat#:M02975 | |
| Chemical compound, drug | Oilgo(dT)$_{20}$ | Invitrogen, Life Technologies | Cat#: 18418020 | |
| Chemical compound, drug | Nuclease Free Water | Invitrogen, Life Technologies | Cat#: AM9937 | |

*Continued on next page*

*Continued*

| Reagent type (species) or resource | Designation | Source or reference | Identifiers | Additional information |
|---|---|---|---|---|
| Chemical compound, drug | Trizol | Invitrogen, Life Technologies | Cat#: 15596026 | |
| Chemical compound, drug | Chloroform | Fisher scientific | Cat#: AC423550250 | |
| Chemical compound, drug | RQ-1 Rnase Free -DNAse | RQ-1 Rnase Free -DNAse | Cat#:M6101 | |
| Software, algorithm | ImageJ | NIH | https://imagej.nih.gov/ij/ | |
| Software, algorithm | FIJI | NIH PMID:22743772 | | |
| Software, algorithm | Adobe Photoshop | Adobe Creative Cloud | N/A | |
| Software, algorithm | Adobe Illustrator | Adobe Creative Cloud | N/A | |
| Software, algorithm | T-COFFEE Alignment Tool | PMID:10964570 | | |
| Software, algorithm | Boxshade | n/a | Swiss Institute of Bioinformatics | Free Open Source Software |
| Software, algorithm | Graphpad PRISM 9 | Graphpad | N/A | |
| Other | ABI7300 | Applied Biosystems | N/A | |
| Other | Nanodrop OneC | ThermoFisher | N/A | |
| Other | LSM 880 Confocal Microscope | Zeiss | N/A | |
| Other | Tangle-Trap | TangleFoot | Cat#:300000588 | |
| Other | RNase-Free Pellet Pestle | VWR | Cat:# 47747–370 | |
| Other | Homogenizer | VWR | Cat:# 749521–1590 | |
| Other | DAPI | Molecular Probes | Cat#: D1306 | (1:1000), 2 hr |
| Other | Phalloidin 647 | Abcam | Cat#: Ab176759 | (1:1000), 2 hr |

## Fly stocks and husbandry

Flies were raised on standard *Drosophila* medium at 25°C. For aging experiments, adult flies were collected each day and raised at 29°C. Flies used in experiments with the presence of TsGal80 (*McGuire et al., 2003*) were raised at 18°C until eclosion, and then shifted to 29.0°C.

The following fly stocks were obtained from the Bloomington *Drosophila* Stock Center: *Oregon-R* (5), Df(3R)ED5938 (24139) (*Ryder et al., 2007*), CG31475[MI08258] (51072) (*Nagarkar-Jaiswal et al., 2015*), CG31475[MI08666] (51100) (*Nagarkar-Jaiswal et al., 2015*), CG31475[MB03509] (24073) (*Bellen et al., 2011*), Tubulin-Gal4-(5138) (*Lee and Luo, 1999*), Elav[C155]-Gal4 (458) (*Lin and Goodman, 1994*), MHC-Gal4 (55132) (*Klein et al., 2014*), MHC-Gal4 (55133) (*Klein et al., 2014*), TsGal80 (7108) (*McGuire et al., 2003*), TsGal80 (7017) (*McGuire et al., 2003*), OK371-Gal4 (26160) (*Mahr and Aberle, 2006*), Repo-Gal4 (7415) (*Sepp et al., 2001*), UAS-Dicer 2 (24651) (*Dietzl et al., 2007*), UAS-Luciferase (35788) (*Perkins et al., 2015*), UAS-Golgi-Galt-RFP (65251) (*Zhou et al., 2014*), Gbb[D20] (63054) (*Chen et al., 1998*), UAS-Tkv[CA] (36537) (*Hoodless et al., 1996*), Tkv[7] (3242) (*Nüsslein-Volhard et al., 1984*), Tkv[8] (34509) (*Nüsslein-Volhard et al., 1984*), Wit[B11] (5174) (*Harrison et al., 1995*), and Mad[1-2] (7323) (*Wiersdorff et al., 1996*). The following stocks were

obtained from the Vienna *Drosophila* Resource Center: UAS-mayday[RNAi] (106664) (*Dietzl et al., 2007*). Other Fly Stocks Include: *Myd[3PM71]* (*Palladino et al., 2002*). Both UAS-Venus::myd and UAS-Cab45 were generated in the Babcock Lab (see below).

## Generation of transgenic fly stocks

UAS-*Venus::myd* was created by amplifying myd cDNA from a Gold cDNA clone (16308) obtained from the *Drosophila* Genomics Resource Center. *Myd* cDNA was first cloned into pCR8 using the pCR8 Gateway cloning kit (ThermoFisher Scientific) and then cloned into the pBID-UASC-VG vector (Addgene #35206 deposited by Brain McCabe) (*Wang et al., 2012*) with the venus tag attached to the N-Terminus of Myd. The construct was inserted into Chromosome two at VK00002 by BestGene Inc (Chino Hills, CA).

UAS-Cab45 was generated using full-length WT Cab45 that was isolated from HeLa cell culture using the Monarch Total RNA Isolation kit (New England Biolabs). Next, total RNA was amplified by RT-PCR using the qScript cDNA Synthesis Kit (Quantabio). Cab45 DNA was then amplified from the cDNA using the following primers: Forward 5'-ATGGTCTGGCCCTGGGTG-3' and Reverse 5'-TCAAAACTCCTCGTGCACGCT-3' (Integrated DNA Technologies) (IDT). Cab45 cDNA was then cloned into pCR8, followed by cloning into pBID-UASC-G vector (Addgene #35202 deposited by Brain McCabe) (*Wang et al., 2012*). The UAS-Cab45 construct was inserted into Chromosome two at VK00002 by BestGene, Inc (Chino Hills, CA). DNA sequencing was performed at each step to verify construct identity (Genscript).

## Immunohistochemistry

Dorsal Longitudinal Muscles (DLMs) were dissected by first removing the head and abdomen from the thorax. Samples were then fixed in 4% paraformaldehyde in PBS for 30 min, and washed four times with PBS at room temperature (RT). Thoraces were then flash frozen with liquid nitrogen and bisected down the midline in ice cold PBS. Tissues were then incubated in blocking buffer (PBS with 0.1% normal goat serum and 0.2% Triton X-100) for at least 1 hr at 4°C (*Sidisky and Babcock, 2020*). The following primary antibodies were used to treat samples overnight at 4°C unless otherwise specified; rabbit polyclonal anti-DsRed 1:500 (Takara Bio USA Inc), chicken polyclonal anti-GFP 1:500 (ThermoFisher), rabbit monoclonal anti-pMad 1:500 (Abcam ab52903) and rat anti-elav 1:20 (Developmental Studies Hybridoma Bank Elav-9F8A9). The following primary antibodies were treated for 48 hr at 4°C, mouse monoclonal anti-Gbb 1:200 (Developmental Studies Hybridoma Bank 3D6-24), rabbit polyclonal anti-Lamp-1 1:500 (Abcam ab30687), and rabbit polyclonal anti-Rab5 1:200 (Abcam ab31261). Tissues were then washed four times in PBS with 0.3% Triton X-100 (PBS-T) for 5 min followed by secondary antibody treatment. The following secondary antibodies were used for 2 hr at RT: species specific Alexa-488 and Alexa 568 (Molecular Probes) were used at 1:200, DAPI 1:1000 (D1306, Invitrogen, Molecular Probes) Cy3-conjugated anti HRP 1:500, FITC-conjugated anti HRP (Jackson Laboratories) 1:200, and Phalloidin iFluor 647 −1:1000 (Abcam ab176759). Samples were washed four times with PBS-T and then mounted on a glass slide with Vectashield (Vector Laboratories).

## TUNEL assay

Adult thoraces were isolated from Day 28 old flies as described above fixed for 30 min at RT, washed 4X with PBS and bisected in ice cold PBS after treatment with liquid nitrogen. Tissue were permeabilized and blocked for 1 hr at 4°C with and washed 4X with PBST for 5 min before TUNEL staining. Thoraces were then treated with the In Situ Death Detection Kit Fluorescein (Roche, Germany). Each sample was treated with 7 μL tdt enzyme and 70 μL Fluorescein Labeling Mix for 3 hr at 37.0°C adapted from *Denton and Kumar, 2015*. Samples were washed 4X with PBST and stained with DAPI 1:1000 in PBST-0.1% NGS for 20 min at RT, washed 4x with PBST and mounted with Vectashield (*Wang et al., 2016*).

Thoracic ganglia from Day 25 old flies were removed from the thorax and fixed in 4% formaldehyde. Tissues were then blocked and stained as described above for DLM tissues.

## Imaging acquisition

A Zeiss LSM 880 Confocal Microscope was used to captured DLM images using a 63 X oil objective (N.A. 1.4). Thoracic ganglion images were obtained using a ×40 oil objective (N.A. 1.3) Confocal stacks were generated using parameters specified under each assay description. Brightness and contrast were adjusted using ImageJ software (NIH) Fiji (*Schindelin et al., 2012*) and Adobe Photoshop CC2020. All figures were generated in Adobe Illustrator CC2020.

## DLM synaptic morphology

Images for DLM for all synaptic morphology were captured using a ×63 objective (N.A. 1.4) oil by creating a Z-Stack at a constant tissue depth of 45 slices (with an interval of 0.7 µm) from the top of the tissue when HRP staining first comes into view at muscle fiber D, as indicated by the placement of the white box in *Figure 1A*. A total of 20 images were captured for each condition with identical parameters for each experiment. Images were then processed as Max Intensity Projections (MIP) using Fiji software (*Schindelin et al., 2012*). Synaptic Morphology Measurements of total neurite length (µM) and branch number were obtained through traces made form HRP staining using the updated Simple Neurite Tracer (SNT) Plug-in (*Arshadi et al., 2020*; *Longair et al., 2011*) and analyzed using the Skeletonize 3D Plug-in of Fiji (*Schindelin et al., 2012*) for each image. Boutons were counted manually for each image using the Cell Counter tool in Fiji (*Schindelin et al., 2012*).

## Quantification of pMad in DLMs and motor neurons

The muscle pMad was quantified using the Analyze Particles Plugin in Fiji. Z-Stacks of 30 slices (an interval of 1.0 µM) in muscle Fiber D were obtained processed as MIPs. Background was subtracted and threshold was adjusted. The pMad puncta were quantified as the percentage of pMad puncta per muscle area for 10 images for each condition.

Thoracic ganglion preps stained with Elav and pMad were used to identify motor neuron nuclei that are pMad positive. Z-stacks of 15 slices (0.7 µM interval) of each lobe (T1 and T2) with using a ×40 oil (N.A 1.3) objective were obtained. The total number of pMad + nuclei per image was analyzed by counting nuclei stained with both Elav and pMad using the Cell Counter tool in Fiji with a total of 12 images per condition.

## Quantification of Gbb

The Gbb in DLM muscle tissue was quantified using the Analyze Particles tool in Fiji. Z-Stacks of 30 slices (interval of 1.0 µM) of muscle Fiber D were obtained and processed as MIPs. Background was subtracted and threshold was adjusted. The Gbb puncta were calculated as the total Gbb puncta per muscle area over 10 images, for each condition.

For the Gbb trafficking images and colocalization analysis, a Z-stack with 15 slices (interval of 0.7 µM) for each image of the same area in muscle Fiber D and processed as a MIP. The colocalized puncta were counted manually using the Cell Counter tool in Fiji in areas where the Gbb puncta clearly overlapped with the marker in each condition per muscle area. For each condition, 10 images were analyzed.

## TUNEL quantification

The percent of TUNEL positive nuclei in DLM tissue was obtained from Z-stacks of 15 slices (interval of 0.7 µM) and processed as an MIP in FIJI. The percent of TUNEL positive nuclei was quantified by manually counting the total number nuclei with DAPI and then count the nuclei that had a TUNEL-positive signal. The percent of TUNEL positive nuclei was tabulated for each image, a total of 10 images were generated and analyzed for each condition from the same area in muscle fiber D.

The number of TUNEL-positive nuclei in the thoracic ganglion were obtained from Z-Stacks of 15 slices (0.7 µM interval) and processed as a MIP in FIJI in each lobe (T1 and T2) with using a ×40 oil (N.A 1.3) objective. The total number of TUNEL + nuclei per image was analyzed counting nuclei stained with both DAPI and TUNEL using the Cell Counter tool in Fiji with a total of 12 images per condition.

## Thorax RNA isolation and qRT-PCR analysis

Total RNA was isolated from Day 28 old flies with 30 adult thoraces isolated per genotype that were dissected in ice cold PBS. PBS was removed prior to snap freezing the tissue with liquid nitrogen and stored at −80℃ for storage until RNA isolation. The RNA was isolated (*Kearse et al., 2011*) by homogenizing tissue in Trizol (Invitrogen) and chloroform following the manufacturer's instructions. Each tissue sample was homogenized using an RNAase-free pestal. RNAase free glycogen (Invitrogen) was used to increase the yield. Total RNA was then washed once with 75% EtOH and with 100.0% EtOH and resuspended in Nuclease-Free water (Invitrogen). Total RNA samples were processed using the NEB RNA Clean-up kit (NEB T2030) to remove any potential residual contaminants from Trizol isolation. RNA quality and concentration were accessed using a Nanodrop One C prior to downstream applications. Total RNA was stored at −80℃.

Total RNA of 1.0 µg was treated with Promega DNase (M6101) following manufacturer's instructions. Total RNA was then reverse transcribed using SuperScript III Reverse Transcriptase (Invitrogen), RNAase Out (Invitrogen) and Oligo(dT)$_{20}$ Primer (Invitrogen) following manufacturer's instructions. cDNA was also subjected to an RNase H (NEB M02975) treatment to remove any mRNA prior to qPCR analysis.

For qRT-PCR analysis was performed using an ABI7300 Real-Time PCR system using SYBR green power up master mix (Applied Biosystems) on cDNA. Actin5C primers (*Dalui and Bhattacharyya, 2014*) Forward 5'- CGAAGAAGTTGCTGCTCTGGTTGT-3' and Reverse 5'- GGACGTCCCACAATCGATGGGAAG-3' were used as an internal control and CG31475 primers (*Dietzl et al., 2007*) Forward 5'- TCCAGGAATTGGGGCAGTACATAAATC-3' and Reverse 5' CTCGGGATGGCGGAAACTCA-3' were used to detect myd. Reactions were repeated in triplicate to obtain average cycle number ($C_t$) values. Actin5C was the internal control. The fold change in myd gene expression was adapted from as previously described (*Bhattacharya et al., 2018*; *Ton and Iovine, 2013*). The delta Ct (▲Ct) values represent expression levels normalized to Actin 5C. The delta delta Ct (▲▲Ct) values represent the relative gene expression levels. The 2$^{-▲▲Ct}$ method was used to calculate the fold change in expression of myd relative to WT across each genotype.

## Flight behavior

Flies were for each genotype were collected, separated by sex, and aged at 29.0℃. Flight behavior was characterized using the flight test as previously described (*Babcock and Ganetzky, 2014*). Briefly, flies were transferred into glass vials and launched down into a 90 cm tube with the inner walls coated with Tangle-Trap (TangleFoot). The landing height of each individual fly was measured to the nearest cm. For each experiment, males and females were tested separately and the landing average was recorded. Data of both sexes was combined for each genotype if no statistical significance was detected between sexes.

## Protein alignment

The protein alignment of Cab45 and Mayday was generated using the Cab45 (NP_057260.2) and *Drosophila* CG31475 (NP_001262725) sequences. The alignment was generated using T-Coffee Alignment tool (*Notredame et al., 2000*). The shading the alignment was generated using the Boxshade software from the Swiss Institute of Bioinformatics (SIB). The EF-hand domains were identified using the UnitProtKB database (*Apweiler et al., 2017 Apweiler, 2004*) for Cab45 (Q9VDY9) and CG31475 (Q9BRK.1).

## Statistical analysis

Statistical analysis for the data was conducted using the Brown-Forsythe and Welch ANOVA tests with Post hoc Dunnett's or Games-Howell multiple comparisons or a one way ANOVA with Turkey Post hoc comparisons or a Student's T-Test where appropriate in Graphpad PRISM 9 (Graphpad Software, San Diego),CA.

## Acknowledgements

The authors thank Robert Kreber for fly stocks, and members of the Babcock laboratory for helpful discussions and suggestions. This research was supported by the National Institutes of Health (R01 NS110727 to DTB).

## Additional information

### Funding

| Funder | Grant reference number | Author |
|---|---|---|
| National Institutes of Health | R01NS110727 | Daniel Babcock |

The funders had no role in study design, data collection and interpretation, or the decision to submit the work for publication.

### Author contributions

Jessica M Sidisky, Conceptualization, Formal analysis, Investigation, Methodology, Writing - original draft, Writing - review and editing; Daniel Weaver, Sarrah Hussain, Investigation, Methodology; Meryem Okumus, Russell Caratenuto, Investigation; Daniel Babcock, Conceptualization, Resources, Formal analysis, Supervision, Funding acquisition, Investigation, Methodology, Writing - original draft, Project administration, Writing - review and editing

### Author ORCIDs

Daniel Babcock ⓘ https://orcid.org/0000-0002-8102-9133

### Decision letter and Author response

Decision letter https://doi.org/10.7554/eLife.54932.sa1
Author response https://doi.org/10.7554/eLife.54932.sa2

## Additional files

### Supplementary files

- Supplementary file 1. Average landing heights and sample sizes for *Figure 4*.
- Supplementary file 2. Average landing heights and sample sizes for *Figure 9*.
- Transparent reporting form

### Data availability

All data generated during this study are included in the manuscript and supporting files.

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
