## [Decision Letter]

**Acceptance summary:**

The authors show that mutations in a gene, mayday, cause flightlessness in 28-day old fruit flies. The function of this gene is necessary for adult neuromuscular junction maintenance. The Mayday protein, and its functional human ortholog Cab45, localizes to Golgi. The secretion of a ligand for BMP signaling, Glass Bottom Boat (GBB) from muscles to motoneurons is mediated by Mayday. In mayday mutants, the defective BMP signaling in neurons feeds back to the muscle to cause muscle degeneration.

**Decision letter after peer review:**

Thank you for submitting your article "Mayday sustains trans-synaptic BMP signaling required for synaptic maintenance with age" for consideration by *eLife*. Your article has been reviewed by two peer reviewers, and the evaluation has been overseen by K VijayRaghavan as the Senior Editor and Reviewing Editor. The reviewers have opted to remain anonymous.

The reviewers have discussed the reviews with one another and the Reviewing Editor has drafted this decision to help you prepare a revised submission.

Summary:

In this study, the authors identified a gene, a mutation in which causes flightlessness in 28-day old fruit flies. They assert that the function of this gene (CG31475, Mayday), is necessary for adult NMJ maintenance, that Mayday, and its functional human ortholog Cab45, localizes to Golgi, that Gbb secretion from DLMs to motoneurons is mediated by Mayday that in the absence of Mayday, GBB secretion from the muscle and therefore, Mad phosphorylation in motoneurons does not occur. This leads to NMJ deterioration, therefore, flightlessness in myd-/- is a consequence of deteriorating NMJs leading to cell death in DLMs.

Overall, this is a nice finding for a poorly characterized protein that is conserved from flies to human. Linking it to the BMP pathway and finding a phenotype in NMJ aging will be interesting for the field. However, there is a lack of quantification throughout the manuscript that needs to be addressed. If the author is resource-limited in some of the common toolkits available to many, there may still be ways to address key issues. That being said, the quantification requested can be done with the current images the authors have in place. Without the quantification of GBB, pMAD and Tunel staining, we're just left with representative images and a crude assay for flight behavior. Adding that quantification and some co-staining for what compartment the GBB is building up in following the loss of Mayday would put the paper over the bar, given its a novel and poorly understood protein. Most other issues can address be addressed in the Discussion. Again, If the data fully supported conclusions being made, which they might with the suggested changes and additional experiments and quantitations, the manuscript would be ready for acceptance. The authors should update the manuscript with additional data and quantitations as listed in the reviews.

Essential revisions:

Portions of the presented data, and their analogy to previous observations with regard to TgF-b signalling in embryonic NMJ function, support the proposed model. Lacunae in the data do not support most points in their assertion (introductory paragraph of this review).

1) NMJ Quantitation: NMJs in adult *Drosophila* DLMs, compared to embryonic NMJs, are numerous and sample preparation is tricky. Single high resolution images, as shown in Figures 1, 3, 5, 6 ,7 that depict NMJs in adult DLMs are inadequate to document the general state of NMJs in DLMs. An illustration of this can be found in Figures 1N, 4C and 6L which are key to the argument regarding NMJ health. For instance in Figure 1N, the appearance of neurites between muscle fibers suggests that image was taken in the depth of tissue, rather than the surface where neurites might be seen more clearly. Further single images as presented here do not describe variations in the observation sufficiently. To make the quantification of NMJ numbers above reproach, we recommend the neurites in the sample area be imaged in tiled 3D stacks of constant depth (30u) with the surface of myofibrils marking the middle slice, imaged at the Nyquist limit, with sufficiently large sample set (at least >10 animals, 20 hemithoraces, per group). The total length of neurites, number of boutons and Scholl analysis on these datasets will convey the authors' observation quantitatively, also revealing the variation induced by technique. These parameters are likely to be good benchmarks for NMJ quantitation. In a future manuscript, at least one representative 3D rendering of these datasets per group should be presented along with the above quantitation. It is possible that this method of quantitation will set a precedence for further such studies. Considering the size of DLMs and the number of NMJs they bear, a detailed quantification of the number of NMJs in the entire hemithorax is impractical, though desirable. If the authors choose to limit their analysis to the inset specified in the schematic in Figure 1A, that would be acceptable.

2) Neurite imaging: Images of neurites presented in this manuscript appear saturated, especially Figures 5, 6 and 7. This must be avoided to discern boutons.

3) Mayday transcript reduction: While the genetic tests over the deficiency suggest that the mutation affects CG31745, a supplemental qRT PCR comparison of transcript levels between the wildtype and 3PM71 alleles is essential.

4) Gbb localization: The imaging data suggest that Gbb secretion from DLMs is reduced and that Gbb accumulates in DLMs. Since the amount of tissue in this context is not limiting, levels of Gbb accumulation in DLMs should be demonstrated through western blots to eliminate the limit in quantitation placed by single images.

5) Presynaptic pMad localization: To demonstrate presynaptic pMad localization (key to the model in Figure 8) the images of boutons need to be of higher resolution and magnification. The magnification used herein, combined with the absence of guiding arrows, make it very difficult to verify the claim in the text. The rabbit anti PS1 antibody (Persson et al.) may work better than the Cell signaling Antibody used for this IHC.

Also, does pMad accumulate in the cell body of the motor neuron as shown in embryos? This experiment will help the authors comment on whether presynaptic pMad localization (as claimed) has a local function or in fact travels all the way to the brain.

6) Mayday function in motor neurons: The fact that the age dependent flight defect in myd -/- animals can be rescued through ubiquitous over expression, but not muscle specific overexpression, begs the question if myd function is required in motor neurons as well. Though myd knockdown in motor neurons causes no flight defect, ascertaining the function of myd in motor neurons a little further will be useful. For this, overexpressing myd with MHC-Gal4 and OK371-Gal4 simultaneously in a myd -/- background will be informative.

7) RNAi usage: The authors employed the RNAi line used in the screen reported in Dietzl et al., 2007. If the specificity of this construct has been previously demonstrated, that study needs to be cited. If not, the specificity of this construct needs to be experimentally demonstrated. Because the knockdown phenotype matches the allele phenotype, the least the authors need to do is to rescue the knockdown phenotype with the UAS-myd transgene. The availability of viable mutant alleles and the ability to overexpress the WT protein in a tissue-specific manner makes RNAi unnecessary in these experiments and should, therefore, be avoided.

8) Muscle function in aged myd -/- animals: The authors demonstrate cell death in aging myd mutants. At a time point prior to complete loss of flight, a functional readout of motor neurons could be shown, if speedily feasible, either through electrophysiology or through genetically encoded calcium indicators.

9) Why is the loss of GBB a maintenance issue versus a developmental issue? Is GBB normally secreted early on, but then fails. Why would cargo sorting in the TGN be independent of Mayday early on in development, but then become dependent on it later? Or is this a case where the GBB pathway itself is more maintenance versus development in general. If you disrupt the GBB pathway only after metamorphosis, do you get the same age-dependent phenotype or is GBB required earlier than Mayday? Any thoughts on this key point in what Mayday is actually doing for GBB signaling.

10) Is there an effect at the larval NMJ where the pathway has been well-described, or only in adults? If not in larvae, why is Mayday only required in adults?

11) What is Mayday/Cab45 actually doing – is it required to form the TGN post-vesicle, or is it leading to a loss of other key proteins that should also be on the vesicle? In this regard, it would be useful to have more staining done for Figure 6 data and added as a supplement. Where is all that extra GBB accumulating? It looks nothing like the Mayday-venus staining in Figure 5, where it co-localizes with a Golgi marker in a compartment right by the nucleus. Do those GBB vesicles co-stain for Golgi markers or other secretory markers (Rabs, SNAREs) or are they targeted to lysosomes and co-stain with lysosomal markers? It would be important to have some insight into where the pathway goes awry in the absence of Mayday. Does GBB fail to load in vesicles at all, or are they the wrong vesicles, or vesicles lacking other key cargo such that they can't fuse?

12) The authors should quantify the GBB, pMad and Tunel staining for their data. They show representative images and indicate its normal or abnormal at different time points, but without any quantification, we don't have data to support that. They only show behavioral flight landing data. Quantification of the other phenotypes is really required.

---

## [Author Response]

Essential revisions:Portions of the presented data, and their analogy to previous observations with regard to TgF-b signalling in embryonic NMJ function, support the proposed model. Lacunae in the data do not support most points in their assertion (introductory paragraph of this review).1) NMJ Quantitation: NMJs in adult *Drosophila* DLMs, compared to embryonic NMJs, are numerous and sample preparation is tricky. Single high resolution images, as shown in Figures 1, 3, 5, 6 ,7 that depict NMJs in adult DLMs are inadequate to document the general state of NMJs in DLMs. An illustration of this can be found in Figures 1N, 4C and 6L which are key to the argument regarding NMJ health. For instance in Figure 1N, the appearance of neurites between muscle fibers suggests that image was taken in the depth of tissue, rather than the surface where neurites might be seen more clearly. Further single images as presented here do not describe variations in the observation sufficiently. To make the quantification of NMJ numbers above reproach, we recommend the neurites in the sample area be imaged in tiled 3D stacks of constant depth (30u) with the surface of myofibrils marking the middle slice, imaged at the Nyquist limit, with sufficiently large sample set (at least >10 animals, 20 hemithoraces, per group). The total length of neurites, number of boutons and Scholl analysis on these datasets will convey the authors' observation quantitatively, also revealing the variation induced by technique. These parameters are likely to be good benchmarks for NMJ quantitation. In a future manuscript, at least one representative 3D rendering of these datasets per group should be presented along with the above quantitation. It is possible that this method of quantitation will set a precedence for further such studies. Considering the size of DLMs and the number of NMJs they bear, a detailed quantification of the number of NMJs in the entire hemithorax is impractical, though desirable. If the authors choose to limit their analysis to the inset specified in the schematic in Figure 1A, that would be acceptable.

We agree that quantification of these adult NMJs is very important not only for this manuscript, but for future studies involving the use of this tissue as well. For all data sets involving NMJ morphology, we have added quantitative measurements to assess the structural integrity of these synapses. These measurements include total branch length, bouton number, and branch number over a specified area. The inclusion of these measurements provides greater clarity on the structural deterioration of these NMJs, and also illustrates these values across a sample size of 20 NMJs per condition.

To standardize the NMJ images throughout the manuscript, we acquired new confocal images on these tissue samples using a consistent starting depth, total depth, and specific muscle area. These parameters are listed in our updated Materials and methods section. These updated images and quantifications can now be found in each figure. Please note the updated figure numbers from our initial submission.

2) Neurite imaging: Images of neurites presented in this manuscript appear saturated, especially Figures 5, 6 and 7. This must be avoided to discern boutons.

Thank you for pointing this out. Since the confocal images were retaken for the values mentioned in the previous comment, we made sure that the images were not saturated in order to discern boutons more clearly. These changes were made for all NMJ images throughout the manuscript.

3) Mayday transcript reduction: While the genetic tests over the deficiency suggest that the mutation affects CG31745, a supplemental qRT PCR comparison of transcript levels between the wildtype and 3PM71 alleles is essential.

We agree that a measurement of CG31475 transcript levels in both 3PM71 mutants and wildtype flies will help to better characterize the 3PM71 mutant allele. We have also measured transcript levels in the publicly available insertion alleles of CG31475. Interestingly, we did not find any significant alterations in transcript levels between WT flies and any of the CG31475 alleles. Rather than reducing transcripts, these results suggest that the CG31475 alleles exert their effects through some other mechanisms; perhaps by interfering with the normal function of the encoded protein. These measurements can be found in Figure 3—figure supplement 1.

4) Gbb localization: The imaging data suggest that Gbb secretion from DLMs is reduced and that Gbb accumulates in DLMs. Since the amount of tissue in this context is not limiting, levels of Gbb accumulation in DLMs should be demonstrated through western blots to eliminate the limit in quantitation placed by single images.

We have included a quantitative assessment of Gbb levels within DLMs and compared them between wildtype and 3PM71 mutants. Specifically, we measured the total amount of Gbb staining over a defined area of muscle tissue in both WT and myd mutant flies. We chose this method of assessment because a western blot would not allow us to separate presynaptic and postsynaptic tissue. These measurements can be found in Figure 8.

5) Presynaptic pMad localization: To demonstrate presynaptic pMad localization (key to the model in Figure 8) the images of boutons need to be of higher resolution and magnification. The magnification used herein, combined with the absence of guiding arrows, make it very difficult to verify the claim in the text. The rabbit anti PS1 antibody (Persson et al.) may work better than the Cell signaling Antibody used for this IHC.Also, does pMad accumulate in the cell body of the motor neuron as shown in embryos? This experiment will help the authors comment on whether presynaptic pMad localization (as claimed) has a local function or in fact travels all the way to the brain.

We have now included quantitative measurements of pMAD staining within motor neuron nuclei located within the thoracic ganglion. By measuring the number of pMAD+ nuclei in both WT and myd mutants, we found a substantial reduction in nuclear pMAD+ signaling in motor neurons in myd mutants. This result shows that the pMAD does indeed travel to the motor neuron nuclei, and is decreased significantly as a result of impair trans-synaptic signaling in myd mutants. These results can be found in an updated Figure 9.

6) Mayday function in motor neurons: The fact that the age dependent flight defect in myd -/- animals can be rescued through ubiquitous over expression, but not muscle specific overexpression, begs the question if myd function is required in motor neurons as well. Though myd knockdown in motor neurons causes no flight defect, ascertaining the function of myd in motor neurons a little further will be useful. For this, overexpressing myd with MHC-Gal4 and OK371-Gal4 simultaneously in a myd -/- background will be informative.

We agree that the inability of any single tissue to rescue the myd -/- mutant phenotype suggests an important role for Myd in multiple tissues. Specifically, myd expression appears to be required in both presynaptic motor neurons as well as postsynaptic muscles to maintain NMJ morphology and flight ability. By expressing UAS-Myd in both muscles and motor neurons in a myd -/- mutant background, as suggested, we find that both flight ability and NMJ integrity are rescued. These results can be found in Figure 5.

7) RNAi usage: The authors employed the RNAi line used in the screen reported in Dietzl et al., 2007. If the specificity of this construct has been previously demonstrated, that study needs to be cited. If not, the specificity of this construct needs to be experimentally demonstrated. Because the knockdown phenotype matches the allele phenotype, the least the authors need to do is to rescue the knockdown phenotype with the UAS-myd transgene. The availability of viable mutant alleles and the ability to overexpress the WT protein in a tissue-specific manner makes RNAi unnecessary in these experiments and should, therefore, be avoided.

To demonstrate the specificity of the RNAi construct, we expressed both UAS-myd-RNAi and UAS-myd::Venus using MHC-Gal4 and found that this rescues both the flight phenotype and NMJ morphology. Additionally we drove expression of both UAS-myd-RNAi and UAS-luciferase to show that this was actually a rescue and not a result of transgene dilution. These results can be found in Figure 4—figure supplement 1.

8) Muscle function in aged myd -/- animals: The authors demonstrate cell death in aging myd mutants. At a time point prior to complete loss of flight, a functional readout of motor neurons could be shown, if speedily feasible, either through electrophysiology or through genetically encoded calcium indicators.

We agree that having a functional readout of motor neurons using electrophysiology will help to provide further insight into the specific defects that occur in myd mutants. We are currently in the process of acquiring equipment for construction of our own electrophysiology rig so that we can make these kinds of measurements. However, setting up the rig and having personnel trained to measure these recordings will be a lengthy process, taking several additional months at least. We did, however, add a statement in the Discussion section mentioning that future studies can further analyze the defects seen in myd mutants using this type of protocol.

9) Why is the loss of GBB a maintenance issue versus a developmental issue? Is GBB normally secreted early on, but then fails. Why would cargo sorting in the TGN be independent of Mayday early on in development, but then become dependent on it later? Or is this a case where the GBB pathway itself is more maintenance versus development in general. If you disrupt the GBB pathway only after metamorphosis, do you get the same age-dependent phenotype or is GBB required earlier than Mayday? Any thoughts on this key point in what Mayday is actually doing for GBB signaling.

The issue of maintenance versus development stems from our results that gross morphology of NMJs as well as flight performance in our mayday mutants are initially similar to their wildtype counterparts in young adult flies. We do not believe it would be accurate to suggest that Mayday is only required in adults, however. Since ubiquitous knockdown of Mayday is lethal (no viable larvae are found), earlier roles are also necessary. The key result is not that Mayday is only required during certain stages of development, but rather that sustained BMP signaling is required beyond development of NMJs.

As suggested, we performed transient knockdown of Mayday only after metamorphosis. Under this condition, we still found a progressive loss of NMJ structure and function, suggesting that this trans-synaptic signaling is required even after the formation of NMJs. These results are found in Figure 4—figure supplement 3.

10) Is there an effect at the larval NMJ where the pathway has been well-described, or only in adults? If not in larvae, why is Mayday only required in adults?

Our results suggest that Mayday is not only required in adults. Since ubiquitous knockdown of myd does not produce viable larvae or adults, it seems that Mayday is also required throughout development. However, the phenotypes shown in 3PM71 mutants as well as with muscle-specific knockdown take several days to manifest, and the transient nature of larval development may not allow enough time for these defects to show up.

11) What is Mayday/Cab45 actually doing – is it required to form the TGN post-vesicle, or is it leading to a loss of other key proteins that should also be on the vesicle? In this regard, it would be useful to have more staining done for Figure 6 data and added as a supplement. Where is all that extra GBB accumulating? It looks nothing like the Mayday-venus staining in Figure 5, where it co-localizes with a Golgi marker in a compartment right by the nucleus. Do those GBB vesicles co-stain for Golgi markers or other secretory markers (Rabs, SNAREs) or are they targeted to lysosomes and co-stain with lysosomal markers? It would be important to have some insight into where the pathway goes awry in the absence of Mayday. Does GBB fail to load in vesicles at all, or are they the wrong vesicles, or vesicles lacking other key cargo such that they can't fuse?

We do find a small (but significant) accumulation of Gbb that colocalize with golgi and endosomal markers in myd mutants. These results appear consistent with items that are normally trafficked. Since the vast majority of Gbb puncta do not appear to co-localize with any membrane-bound components that we examined, however, it is certainly possible that most of the Gbb that accumulates within muscles lies within the cytoplasm. These images and quantification can be found in the updated Figure 8.

12) The authors should quantify the GBB, pMad and Tunel staining for their data. They show representative images and indicate its normal or abnormal at different time points, but without any quantification, we don't have data to support that. They only show behavioral flight landing data. Quantification of the other phenotypes is really required.

We agree with the importance of quantifying the Gbb, pMad, and Tunel staining used throughout the manuscript. We have added measurements of this staining in both DLM and Thoracic Ganglion tissue samples. These results can be found in Figures 8, 9, and 10.